



# An inverse method to relate organic carbon reactivity to isotope composition from serial oxidation

Jordon D. Hemingway[1,2,*], Daniel H. Rothman[3], Sarah Z. Rosengard[1,2,**], and Valier V. Galy[1]

[1]Department of Marine Chemistry and Geochemistry, Woods Hole Oceanographic Institution, 266 Woods Hole Road, Woods Hole, MA 02543, USA

[2]Massachusetts Institute of Technology - Woods Hole Oceanographic Institution Joint Program in Oceanography and Applied Ocean Science and Engineering, 77 Massachusetts Avenue, Cambridge, MA 02139, USA

[3]Lorenz Center, Department of Earth, Atmospheric, and Planetary Science, Massachusetts Institute of Technology, 77 Massachusetts Avenue, Cambridge, MA 02139, USA

[*]Present address: Department of Earth and Planetary Sciences, Harvard University, 20 Oxford Street, Cambridge, MA 02138, USA

[**]Present address: Departments of Geography and Earth, Ocean, and Atmospheric Sciences, University of British Columbia, 2207 Main Mall, Vancouver, BC V6T 1Z4, Canada

*Correspondence to:* Jordon D. Hemingway (jordon_hemingway@fas.harvard.edu)

**Abstract.**

Serial oxidation coupled with stable carbon and radiocarbon analysis of sequentially evolved $CO_2$ is a promising method to characterize the relationship between organic carbon (OC) chemical composition, source, and residence time in the environment. However, observed decay profiles depend on experimental conditions and oxidation pathway. It is therefore necessary to properly assess serial oxidation kinetics before utilizing decay profiles as a measure of OC reactivity. We present a regularized inverse method to estimate the distribution of OC activation energy ($E$), a proxy for bond strength, from ramped temperature pyrolysis/oxidation (RPO). This method directly compares reactivity to isotope composition by determining the $E$ range for OC decaying within each temperature interval over which $CO_2$ is collected. By analyzing a decarbonated test sample at multiple masses and oven ramp rates, we show that OC decay during RPO analysis follows a superposition of parallel first-order kinetics and that resulting $E$ distributions are independent of experimental conditions. We therefore propose the $E$ distribution as a novel proxy to describe OC reactivity and suggest that $E$ vs. isotope relationships can provide new insight into the compositional controls on OC source and residence time. This manuscript is accompanied by an open-source Python package for performing all analyses.

## 1 Introduction

Natural organic matter present in aquatic environments, sediments, soils, and vegetation contains roughly three-fold more carbon than the pre-industrial atmosphere (Bianchi, 2011). As such, the balance between organic carbon (OC) synthesis and



remineralization exerts a major control on the global carbon cycle and on atmospheric $CO_2$ levels (Lasaga et al., 1985). However, OC remineralization rates are spatiotemporally heterogeneous, leading to decay timescales that range from minutes to millions of years (Boudreau and Ruddick, 1991; Forney and Rothman, 2012a; Middelburg, 1989). To explain this variability, it has been hypothesized that remineralization depends on multiple chemical and environmental factors such as OC molecular structure (Burdige, 2007; Tegelaar et al., 1989), microbial community composition (Pedler et al., 2014; Schmidt et al., 2011), secondary chemical interactions (Schmidt et al., 2011), and physical protection by particles (Mikutta et al., 2006; Keil and Mayer, 2014). The relative importance of these governing mechanisms remains actively debated and is thought to vary depending on environmental setting (Hedges et al., 2001; Rothman and Forney, 2007; Schmidt et al., 2011), thus limiting our mechanistic understanding of OC decay.

This limitation is partially methodological in nature; traditional geochemical analyses often target either "bulk" OC or trace "biomarker" molecules such as plant-wax fatty acids (Galy et al., 2011; Galy and Eglinton, 2011; Hemingway et al., 2016). While bulk measurements include all OC contained within a sample, they offer no information on the distribution of chemical structure or reactivity within a complex mixture. In contrast, biomarker analysis is highly specific but individual compounds nonetheless still represent the average of multiple sources. Furthermore, biomarkers typically constitute $\leq 1\%$ of total OC and can be subject to production, transport, and preservation biases (Hemingway et al., 2016).

To bridge the information gained by these methods, a novel class of analytical techniques, termed "serial oxidation," has emerged. Such analyses separate carbon within a bulk sample based on its susceptibility to decomposition by chemical hydrolysis (Helfrich et al., 2007), *uv* light (Beaupré et al., 2007; Follett et al., 2014), heat (Rosenheim et al., 2008), or microbial respiration (Beaupré et al., 2016) and measure the stable carbon ($^{13}C/^{12}C$, expressed as $\delta^{13}C$) and radiocarbon ($^{14}C/^{12}C$, here expressed as fraction modern or Fm) content of evolved $CO_2$. By separating $CO_2$ into multiple lability intervals, isotope ratios are obtained for carbon atoms exhibiting similar physical and/or chemical properties. Because $\delta^{13}C$ provides information on the source of organic matter while Fm reflects the amount of time that has passed since organic compounds were initially synthsized, serial oxidation is a promising method to directly probe the compositional controls on OC source and residence time.

Still, a theoretical treatment of serial oxidation kinetics is lacking, hindering our ability to correlate measured isotope distributions with intrinsic chemical properties and reactivity. In this study, we relate OC thermal recalcitrance to its corresponding $\delta^{13}C$ and Fm values using ramped-temperature pyrolysis/oxidation (RPO). This method involves heating OC at a controlled rate while continuously quantifying and collecting evolved $CO_2$, which is binned over user-defined time intervals (termed "fractions") and analyzed for $\delta^{13}C$ and Fm (Rosenheim et al., 2008; Hemingway et al., 2017). We describe non-isothermal OC decay rates as a function of $E$, the Arrhenius activation energy, using a novel inverse solution to the distributed activation energy model (Braun and Burnham, 1987; Burnham and Braun, 1999; Cramer, 2004; White et al., 2011). By conducting a set of kinetic experiments, we show that the $E$ distribution within a given OC mixture does not depend on experimental conditions and is thus a reliable proxy for bond strength and OC chemical composition.

We begin in Section 3 by deriving the governing equations to describe a parallel superposition of first-order, non-isothermal decay. Then, in Section 4, we describe a method to solve for the distribution of $E$ using a regularized inverse approach.





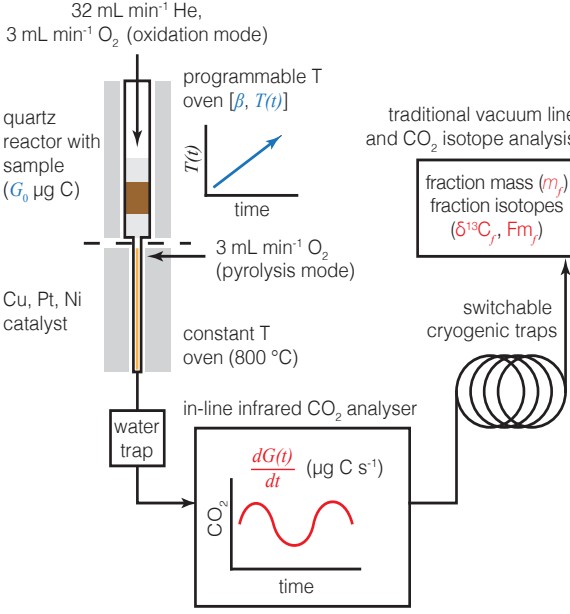

**Figure 1.** RPO instrument schematic. User-defined inputs are printed in blue, while observed measurements are printed in red (See Table 1 for symbol definitions).

Finally, in Section 5, we determine the subset of $E$ that is contained within each RPO fraction and directly relate OC reaction energetics to corresponding isotope values. All calculations were performed using the accompanying 'rampedpyrox' Python package (Hemingway, 2017).

## 2 Materials and Methods

### 2.1 Sample selection and preparation

As a representative sample, we analyzed particulate organic carbon (POC) contained in suspended sediments from the surface of the Narayani River. This sample (PB-60) was collected at the base of the Himalayas (27.70° N, 84.43° E) and has been analyzed for bulk OC and plant-wax carbon isotopes (Galy et al., 2008; Galy and Eglinton, 2011; Galy et al., 2011). Aliquots were taken for RPO analysis from freeze-dried, archived material and acidified under HCl fumes at $60\,°C$ for $72\,h$ to remove carbonates (Whiteside et al., 2011). Because residual chloride has been shown to interact with the RPO catalyst wire (Hemingway et al., 2017), acidified aliquots were rinsed $3\times$ in $18.2\,M\Omega$ MilliQ water and freeze-dried overnight at $-40\,°C$ prior to analysis. For consistency and to properly calculate RPO isotope mass balance, bulk %OC, $\delta^{13}C$, and Fm values were re-measured using fumigated and rinsed material (McNichol et al., 1994a, b). Resulting Fm for rinsed material is 0.04 lower



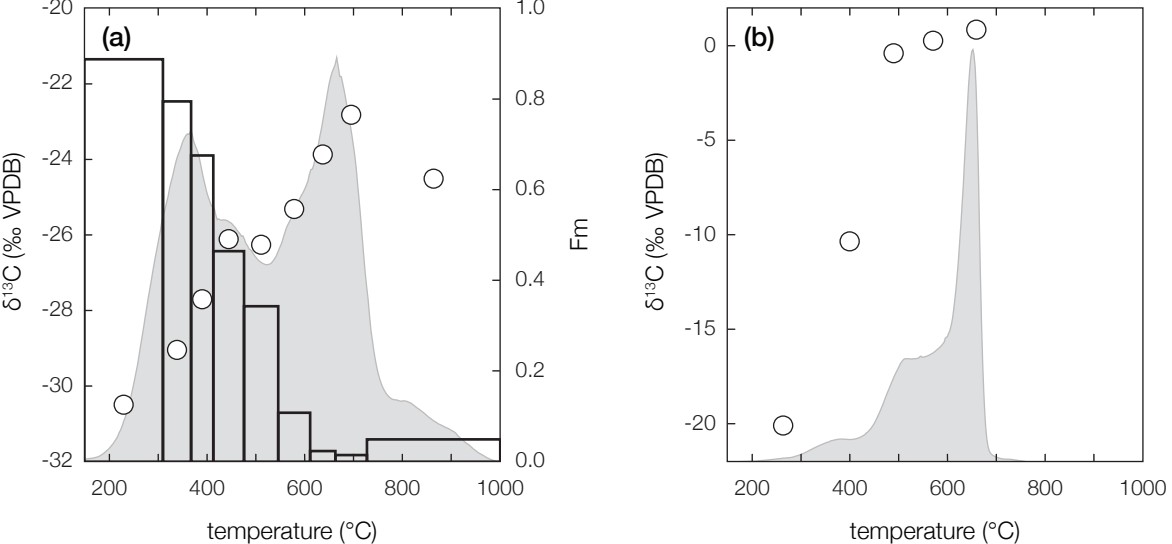

**Figure 2.** RPO results. Mass-normalized thermograms (gray shaded region, unitless), $\delta^{13}C$ values (white circles, left axes), and Fm values (transparent bars, right axes) for **(a)** Narayani POC and **(b)** JGOFS sediment (Fm not measured). Fm bar widths correspond to the temperature range of collection for each RPO fraction. Fm and $\delta^{13}C$ analytical uncertainty is always smaller than point marker and is therefore not shown (see Tables 2–3 for values).

than that for un-rinsed aliquots (Galy et al., 2008), reflecting a minor loss of acid soluble OC for this sample during the rinsing step.

To test if the presence of inorganic carbon (IC; *e.g.* CaCO$_3$) affects decay kinetics, we analyzed a pure CaCO$_3$ laboratory working standard (Icelandic spar; Hemingway et al., 2017) as well as carbonate-rich sediment from the Southern Ocean (60.24°
5 S, 170.19° W) collected for the Joint Global Ocean Flux Study (JGOFS; Sayles et al., 2001). JGOFS aliquots were taken from archived core-top material (0 cm to 0.5 cm, stored at −80 °C), freeze-dried overnight at −40 °C, and homogenized prior to RPO analysis. IC content, OC content, and bulk $^{13}$C composition were re-quantified at NOSAMS (McNichol et al., 1994a).

## 2.2 Instrumental setup

RPO analysis has been described in detail previously (Rosenheim et al., 2008; Hemingway et al., 2017). In summary, a solid
10 sample containing ≈150 µg C to 250 µg C is loaded into a pre-combusted (850 °C, 5 h) quartz reactor and placed into a two-stage oven, as shown in Fig. 1. The reactor is then sealed and the sample is exposed to an atmosphere of 92:8 He:O$_2$ with a flow rate of 35 mL min$^{-1}$ (oxidation mode). During analysis, the oven surrounding the sample is programmed to heat at a user-defined ramp rate, termed $\beta$ (see Table 1 for symbol descriptions). Instantaneous temperature within the oven is measured using two thermocouples separated by ≈1 cm to monitor temperature heterogeneity, which is typically <5 °C. Following standard





practice (Rosenheim et al., 2008), a ramp rate of $5\,°C\,min^{-1}$ was used for all experiments in which $CO_2$ gas was collected for isotope analysis. In the second (downstream) oven, eluent gas is passed over a Cu, Pt, and Ni catalyst wire held at $800\,°C$ to facilitate oxidation of reduced carbon-containing gases to $CO_2$.

After exiting the second oven, eluent gas is distilled through a water trap and passed into a flow-through infrared gas analyzer
(IRGA) to measure $CO_2$ concentration (in parts per million by volume; $ppm\,CO_2$) with 1-s temporal resolution. Resulting $ppm\,CO_2$ vs. temperature plots are referred to as "thermograms" (Fig. 2). IRGA measurements were calibrated using a two-point calibration curve before each analysis to account for instrument drift and are precise to $\pm5\,ppm\,CO_2$ (Hemingway et al., 2017). Downstream of the IRGA, eluent gas is passed into one of two switchable traps and $CO_2$ is cryogenically frozen while He and $O_2$ are vented to the atmosphere. Traps are switched at user-defined time points and $CO_2$ is further distilled, quantified,
transferred into glass tubes packed with $\approx100\,mg\,CuO$ and $\approx10\,mg\,Ag$, and flame sealed. Finally, $CO_2$ was recombusted at $525\,°C$ for 1 h to remove trace contaminant gases.

## 2.3 Isotope measurement, blank correction, and data analysis

Radiocarbon compositions of all bulk samples and RPO fractions were measured at NOSAMS following standard graphitization methods (McNichol et al., 1994b). All radiocarbon results are expressed in fraction modern notation (Fm). We note that
Fm used here is corrected for $^{13}C$ fractionation and is thus identical to the $F^{14}C$ notation of Reimer et al. (2004). Bulk and RPO fraction stable carbon isotope compositions were measured on $CO_2$ gas using a dual-inlet IRMS located at NOSAMS (McNichol et al., 1994a), with resulting $^{13}C$ content expressed in $\delta^{13}C$ per mille (‰) notation relative to Vienna Pee Dee Belemnite (VPDB). RPO fraction masses, $\delta^{13}C$ values, and Fm values were corrected for blank carbon contribution, and $\delta^{13}C$ was additionally corrected to ensure $^{13}C$ mass balance as incomplete oxidation to $CO_2$ has been shown to impart a small fractionation
effect (Hemingway et al., 2017). Analytical uncertainty was propagated throughout all corrections. Thermograms and isotope results for both Narayani POC and JGOFS sediment are plotted in Fig. 2, while temperature ranges, carbon masses, and isotope values are additionally reported in Tables 2–3.

## 3 Deriving a model of decay kinetics

We derive the distributed activation energy model by first considering the case where OC is separated into a set of discrete
components with unique $E$ values. We then generalize this description to allow for a continuous $E$ distribution (Braun and Burnham, 1987; Burnham and Braun, 1999; Cramer, 2004).

### 3.1 Discrete model

During OC remineralization, the decay rate of carbon contained in a particular component $i$ is often described as as a first-order process with respect to $g_i(t)$, the mass of carbon remaining in component $i$ at any time $t$ (Westrich and Berner, 1984; Braun



and Burnham, 1987), according to

$$\frac{dg_i(t)}{dt} = -k_i g_i(t), \tag{1}$$

where $k_i$ is the first-order rate coefficient associated with component $i$. Total OC decay is then treated as the sum over all

components. Although it is possible that OC decay in the natural environment additionally depends on oxidant concentration, we omit this dependency here since $O_2$ is provided in excess in our experimental setup (Fig. 1). In contrast to the "multi-G" and "reactive continuum" models that are often used to describe environmental OC degradation rates (Westrich and Berner, 1984; Boudreau and Ruddick, 1991; Forney and Rothman, 2012a, b), here we allow $k_i$ to vary with time. Because rate coefficients are related to temperature and activation energy, $k_i$ can be written as a time-dependent function of $E$ following the Arrhenius equation:

$$k_i(t) = \omega \exp\left[-\frac{E_i}{RT(t)}\right], \tag{2}$$

where $\omega$ is the empirically derived Arrhenius pre-exponential ("frequency") factor, $R$ is the ideal gas constant, $E_i$ is the activation energy of carbon contained in component $i$, and $T(t)$ is the measured temperature of the system at time $t$. For non-isothermal systems, time-dependent decay coefficients can therefore be described by the static property $E_i$ and the observed variable $T(t)$. Although $T(t)$ is related to $t$ by a constant ramp rate $\beta$ during RPO analysis, we leave this written as is to

emphasize that our model is valid for any measured time-temperature history. Substituting Eq. (2) into Eq. (1), we write the first-order decay at time $t$ during a non-isothermal process as

$$\frac{dg_i(t)}{dt} = -\omega \exp\left[-\frac{E_i}{RT(t)}\right] g_i(t). \tag{3}$$

The mass of carbon remaining in component $i$ at time $t$ can be determined by integrating Eq. (3) from an initial time $t = 0$:

$$g_i(t) = g_i(0) e^{-\kappa_i(t)}, \tag{4}$$

where

$$\kappa_i(t) = \omega \int_0^t \exp\left[-\frac{E_i}{RT(t')}\right] dt' \tag{5}$$

is the time integrated decay coefficient at time $t$ and $g_i(0)$ is the initial mass of carbon contained in component $i$. Equation (5) states that $g_i(t)$ depends on the entire time-temperature history of the experiment. That is, the evolution of $dg_i(t)/dt$ reflects both a decrease in $g_i(t)$ as OC is remineralized and an increase in $k_i(t)$ with increasing $T(t)$ as the experiment progresses.

This results in a predictable shift in RPO thermograms toward higher elution temperatures with increasing $\beta$ (Miura and Maki, 1998), as shown in Fig. 3.

Following Boudreau (1997) and Westrich and Berner (1984), an environmental sample containing a complex OC mixture can be described as a superposition of a finite set of $n$ components, each decaying according to a unique $k_i(t)$ and thus





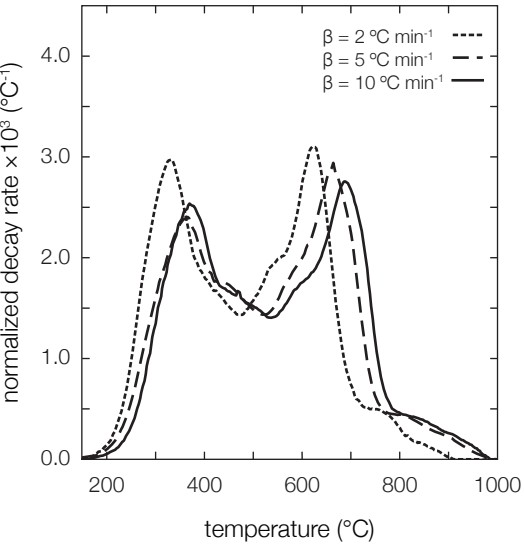

**Figure 3.** Testing the ramp-rate effect. Measured thermograms are shown for Narayani POC analyzed using multiple ramp rates ($\beta$). Decay rates have been normalized by $G_0$ and $\beta$ in order to properly compare $y$ axis values between each analysis.

corresponding to a unique $E_i$ value. $G(t)$, the total carbon mass remaining at $t$, is then the sum of the mass remaining in each component:

$$G(t) = \sum_{i=1}^{n} g_i(t). \tag{6}$$

Substituting Eq. (4) into Eq. (6), this can be written as

5  $$G(t) = \sum_{i=1}^{n} g_i(0)e^{-\kappa_i(t)}. \tag{7}$$

We then define $G_0$, the initial OC mass present in the entire sample, as the sum of initial mass contained in each component:

$$G_0 = \sum_{i=1}^{n} g_i(0). \tag{8}$$

Finally, we define $p_i(0)$, the fraction of total carbon initially contained in component $i$, as

$$p_i(0) = \frac{g_i(0)}{G_0} \tag{9}$$

10  and note that

$$\sum_{i=1}^{n} p_i(0) = 1. \tag{10}$$




Substituting Eq. (9) into Eq. (7) yields

$$\frac{G(t)}{G_0} = \sum_{i=1}^{n} p_i(0) e^{-\kappa_i(t)}, \tag{11}$$

which describes the evolution of the fraction of initial carbon remaining at any time. The fraction of OC initially present within each component, $p_i(0)$, can be determined by fitting Eq. (11) to the observed $G(t)$ profile measured by the RPO instrument.

While informative, this discrete description of the model suffers from two major limitations: (i) $n$ must be set *a priori* or determined empirically (Boudreau and Ruddick, 1991) and (ii) any noise recorded in the data will result in large uncertainty in best-fit $p_i(0)$ and $E_i$ values (Forney and Rothman, 2012b). To circumvent the first of these issues, we derive a more general description of non-isothermal first-order decay that does not assume a finite set of components with unique $E_i$, but rather allows $E$ to vary continuously (Boudreau, 1997; Braun and Burnham, 1987; Burnham and Braun, 1999; Cramer, 2004). The

second problem is then solved using Tikhonov regularization (Section 4.2; Forney and Rothman, 2012b; Hansen, 1994).

## 3.2   Continuous model

In the continuous model, discrete components $g_i(t)$, $\kappa_i(t)$ and $E_i$ are replaced by continuous variables $g(t, E)$, $\kappa(t, E)$ and $E$, respectively (Table 1). Analogous to Eq. (3), we calculate the decay of OC associated with an infinitesimal range $dE$ about any non-negative value of $E$ following first-order Arrhenius kinetics as

$$\frac{dg(t, E)}{dt} = -\omega \exp\left[-\frac{E}{RT(t)}\right] g(t, E). \tag{12}$$

The mass of carbon associated with any value of $E$ that remains unreacted at time $t$ is then calculated by integrating Eq. (12) to obtain

$$g(t, E) = g(0, E) e^{-\kappa(t, E)}, \tag{13}$$

where $g(0, E)$ is the initial mass of carbon associated with activation energy value $E$ and

$$\kappa(t, E) = \omega \int_{0}^{t} \exp\left[-\frac{E}{RT(t')}\right] dt'. \tag{14}$$

The total carbon remaining at time $t$ can now be written as the integral of $g(t, E)$ over all possible values of $E$ as

$$G(t) = \int_{0}^{\infty} g(t, E) dE. \tag{15}$$

Substituting Eq. (13) into Eq. (15), we obtain

$$G(t) = \int_{0}^{\infty} g(0, E) e^{-\kappa(t, E)} dE. \tag{16}$$




Analogous to Eq. (9), we then define the fraction of total carbon initially associated with any value of $E$ as

$$p(0, E) = \frac{g(0, E)}{G_0} \tag{17}$$

where

$$\int_0^\infty p(0, E) dE = 1. \tag{18}$$

Substituting Eq. (17) into Eq. (16) yields:

$$\frac{G(t)}{G_0} = \int_0^\infty p(0, E) e^{-\kappa(t, E)} dE. \tag{19}$$

The distribution of $p(0, E)$ over all values of $E$ describes the initial probability density function (pdf) of $E$ that will lead to the observed OC decay profile when a sample is analyzed in the RPO instrument. As RPO analysis proceeds, this pdf must evolve with time to reflect the fact that some carbon has been remineralized to $CO_2$. Like $g(t, E)$, $p(t, E)$ follows first-order Arrhenius

kinetics according to

$$\frac{dp(t, E)}{dt} = -\omega \exp\left[-\frac{E}{RT(t)}\right] p(t, E), \tag{20}$$

where $p(t, E)$ is the fraction of initial carbon mass that remains associated with $E$ at time $t$. This can be obtained by integrating Eq. (20) from an initial time $t = 0$:

$$p(t, E) = p(0, E) e^{-\kappa(t, E)}. \tag{21}$$

This implies that the carbon initially remineralized to $CO_2$ must be associated with the lowest $E$ values, as low $E$ will lead to a $\kappa(t, E)$ term in Eq. (21) that approaches zero most rapidly. Put differently, OC that is described by higher $E$ values will resist remineralization until more time has passed and, therefore, higher temperatures have been reached – *i.e.* it is more thermally recalcitrant.

### 3.3 First-order verification

Because our model is a specific case of $n^{\text{th}}$-order non-isothermal kinetic models (Braun and Burnham, 1987; White et al., 2011), we must verify that carbon degradation within the RPO instrument behaves according to a superposition of parallel first-order reactions rather than higher-order processes. By replacing $g(t, E)$ with $G_0 p(t, E)$ on the right-hand side of Eq. (12), it can be seen that

$$\frac{dg(t, E)}{dt} = -G_0 \omega \exp\left[-\frac{E}{RT(t)}\right] p(t, E). \tag{22}$$

By integrating over all non-negative values of $E$ and utilizing the definition of $G(t)$ from Eq. (15), this can be written as

$$\frac{dG(t)}{dt} = -G_0 \int_0^\infty \omega \exp\left[-\frac{E}{RT(t)}\right] p(t, E) dE. \tag{23}$$





The first-order model describes $dG(t)/dt$ as a linear function of $G_0$ multiplied by an integral term that depends on $p(t, E)$ but is independent of $G_0$. In contrast, if carbon decomposition within the RPO instrument were to follow a higher-order process, then the relationship between $dG(t)/dt$ and $G_0$ would be nonlinear and would evolve as a function of time (Follett et al., 2014). If we define

$$m(t) = \int_0^\infty \omega \exp\left[-\frac{E}{RT(t)}\right] p(t, E) dE, \qquad (24)$$

then the carbon decay at time $t$ as predicted by parallel first-order kinetics simplifies to

$$\frac{dG(t)}{dt} = -G_0 m(t). \qquad (25)$$

Therefore, similar to the isothermal case (Follett et al., 2014), a superposition of parallel first-order decay reactions will result in a linear relationship between $dG(t)/dt$ and $G_0$ with a zero intercept and a time-dependent slope. Thus, $m(t)$ can be interpreted
as the $G_0$-normalized OC decay rate at time $t$.

    We verify that OC remineralization within the RPO instrument follows parallel first-order kinetics by assessing the linearity between Narayani POC $dG(t)/dt$ and $G_0$ at any time $t$ across a range of $G_0$ values. For 4 arbitrarily chosen time points, this relationship is linear with an ordinary least squares $R^2 \geq 0.999$, resulting in identical $G_0$-normalized thermograms within analytical uncertainty (Fig. 4a–b). Thus, the decay of complex OC mixtures contained in carbonate-free samples during RPO
analysis can indeed be accurately described as a superposition of parallel first-order reactions.

### 3.4   A note of caution on carbonates

While most RPO studies to date have isolated OC by acidifying to remove carbonates (*e.g.* Rosenheim et al., 2008; Rosenheim and Galy, 2012; Rosenheim et al., 2013; Schreiner et al., 2014; Bianchi et al., 2015), it has been argued that acid hydrolysis and/or dissolution of short-range-order minerals during acid treatment can alter the OC chemical bonding environment and
therefore affect thermal stability (Plante et al., 2013). While analyzing samples without acid treatment can circumvent these issues, the effect of carbonates on decay kinetics has not yet been considered. To test if carbonate-rich samples follow parallel first-order kinetics, we analyzed JGOFS sediment for a range of $G_0$ values (Fig. 4c–d). Prior to $t \approx 4500$ s, when $\delta^{13}$C values of eluted $CO_2$ indicate a predominantly OC source (Table 3; Fig. 2b), $dG(t)/dt$ can be accurately described as a linear function of $G_0$ ($R^2 \geq 0.999$). However, as carbonate begins to decompose above $t \approx 4500$ s, the relationship between $dG(t)/dt$ and $G_0$
becomes nonlinear and the carbonate peak shifts toward higher $t$ with increasing $G_0$ (Fig. 4d).

    To investigate if non-first-order decomposition is an intrinsic property of $CaCO_3$ or if this is due to interactions with other materials within the sample (so-called "matrix effects"), we analyzed a pure Icelandic spar $CaCO_3$ labroatory standard at multiple masses ($G_0 = 258\,\mu g\,C$, $492\,\mu g\,C$ and $1014\,\mu g\,C$; $\beta = 5\,°C\,min^{-1}$; not shown). Results indicate that pure carbonate, unlike JGOFS sediment, does follow first-order kinetics with a maximum decomposition rate occurring at $(700 \pm 6)\,°C$ independent
of $G_0$. Interaction with reduced organic carbon, corresponding hetero-atoms (*e.g.* N, P, S), or trace metals contained within the sample matrix are therefore the likeliest cause of non-first-order $CaCO_3$ decomposition when analyzing environmental





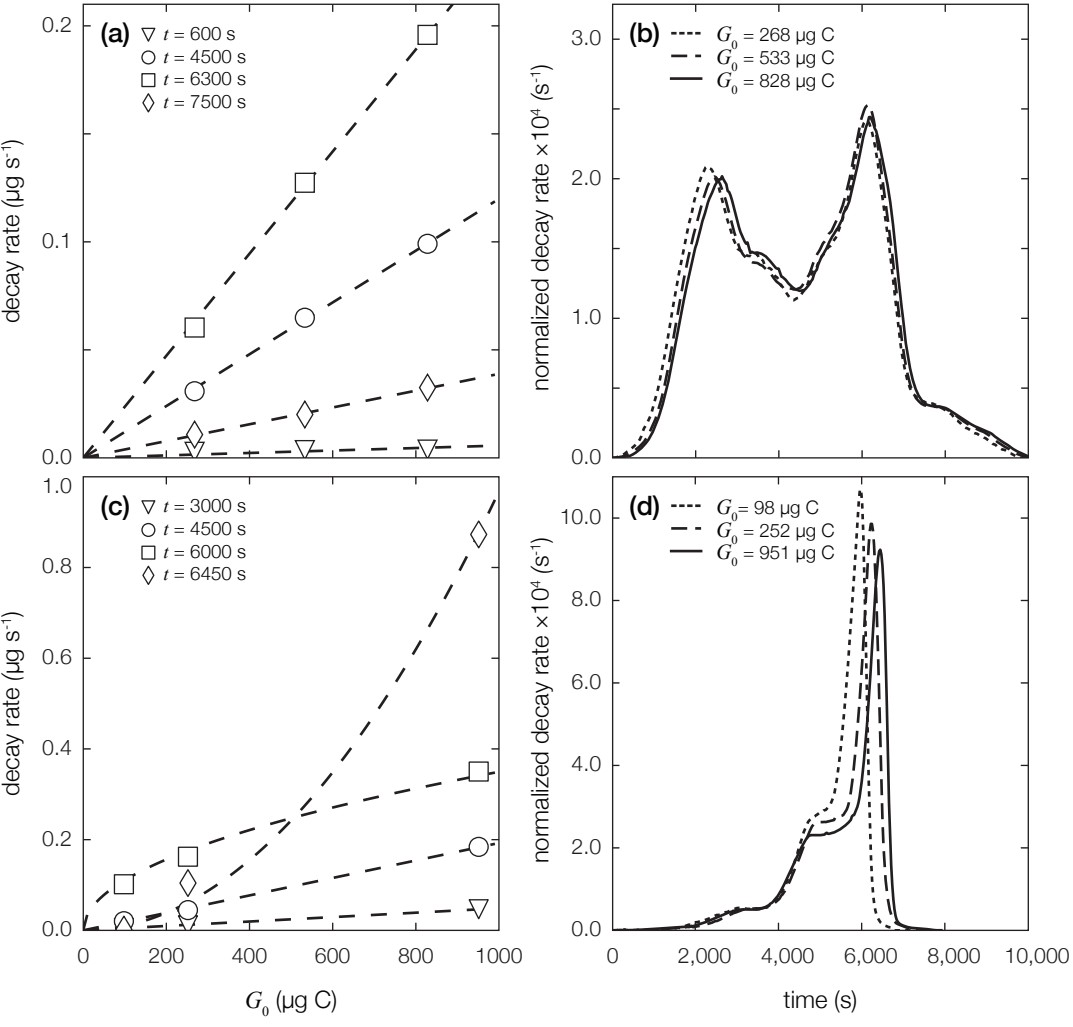

**Figure 4.** First-order kinetic assessment. Left column shows decay rate, $dG(t)/dt$, vs. $G_0$ relationships at four arbitrarily chosen time points (including best-fit regression lines, dashed lines) and right column shows the mass-normalized decay rates [termed $m(t)$ in Eq. (24)–(25)] at all time points for **(a)**–**(b)** Narayani POC and **(c)**–**(d)** JGOFS sediment. Linear relationships and nearly identical normalized decay rates in panels (a)–(b) confirm the first-order nature of OC decay, while non-linear relationships and a shifting carbonate peak in panels (c)–(d) indicate non-first-order $CaCO_3$ decay kinetics.

samples. Thus, while avoiding the issues of acid treatment, the presence of carbonate will result in thermograms that cannot be accurately described by the model presented here, and we therefore argue in favor of acid treatment when using the RPO instrument to determine reaction energetics of carbonate-containing samples.



## 4   Finding the regularized inverse solution

Following Forney and Rothman (2012a, b), we present a method to estimate $p(0, E)$ by finding an inverse solution to Eq. (19). In contrast to previous solutions (Braun and Burnham, 1987; Burnham and Braun, 1999; Cramer, 2004), this approach does not require an *a priori* assumption about the form of $p(0, E)$ (*e.g.* Gaussian). Because this problem is sensitive to noise at the

level of our analytical uncertainty (Forney and Rothman, 2012b), we seek a smooth solution using Tikhonov regularization (Section 4.2; Forney and Rothman, 2012b; Hansen, 1994).

To numerically calculate $p(0, E)$, we discretize the continuous variable $t$ over the time course of the experiment into a vector $\mathbf{t}$ containing $n_t$ nodes such that

$$\Delta t_j = \frac{1}{2} \left( t_{j+1} - t_{j-1} \right), \quad j = 2, \ldots, n_t - 1. \tag{26}$$

For $j = 1$ and $j = n_t$, $t_{j-1}$ and $t_{j+1}$ in Eq. (26) are, respectively, replaced by $t_j$ since $\mathbf{t}$ is undefined outside of this range. For generality, and because the distributed activation energy model is frequently applied over geologic timescales with non-uniformly distributed time measurements (Braun and Burnham, 1987; Burnham and Braun, 1999; Cramer, 2004), Eq. (26) does not require a uniform time step (*i.e.* it is possible that $\Delta t_j \neq \Delta t_{i \neq j}$). Similarly, we generate a vector $\mathbf{E}$ containing $n_E$ nodes over the range values considered for the model solution such that

$$\Delta E = \frac{E_{\max} - E_{\min}}{n_E}, \tag{27}$$

noting that $\mathbf{E}$ is uniformly spaced since this vector is not constrained by observations. We constrain $E$ to be within $50 \, \mathrm{kJ \, mol^{-1}}$ to $350 \, \mathrm{kJ \, mol^{-1}}$ based on published biomass and petroleum $E$ ranges (Braun and Burnham, 1987; Burnham and Braun, 1999; Cramer, 2004; White et al., 2011).

It can be seen from Eq. (19) that our model can be separated into two components: (i) $p(0, E)$ and (ii) the exponentiated, time

integrated decay coefficient, $\exp[\kappa(t, E)]$. Analogous to the Laplace transform for the isothermal reactive continuum model (Forney and Rothman, 2012b), $\exp[\kappa(t, E)]$ determines the fraction of carbon initially associated with an activation energy value $E$ that has decayed by time $t$. While this integral can be calculated analytically for a constant ramp rate $\beta$, here we approximate the solution numerically so that our model can be applied to any time-temperature history. Thus, we populate a matrix $\mathbf{A}$ by calculating $\exp[\kappa(t, E)]$ for each $t_j$ and $E_l$ contained in $\mathbf{t}$ and $\mathbf{E}$ as

$$A_{j,l} = \exp \left\{ -\sum_{u=1}^{j} \omega \exp \left[ -\frac{E_l}{RT(t_u)} \right] \Delta t_u \right\} \Delta E,$$

$$j = 1, \ldots, n_t,$$

$$l = 1, \ldots, n_E. \tag{28}$$

The $\mathbf{A}$ matrix is often termed the model "design matrix." We then calculate the fraction of initial carbon remaining at each time point as

$$\frac{G(t)}{G_0} = 1 - \alpha(t), \tag{29}$$





where $\alpha(t)$ is the $G_0$-normalized, integrated RPO thermogram at time $t$. We generate a discretized vector $\mathbf{g}$ by interpolating $G(t)/G_0$ onto each $t_j$ in $\mathbf{t}$ ($j = 1, \ldots, n_t$). Our model can then be written in matrix form as

$$\mathbf{g} = \mathbf{A} \cdot \mathbf{p}, \tag{30}$$

where $\mathbf{p}$ is an unknown, discretized vector of $p(0, E)$ with length $n_E$ such that

$$p_l = \frac{1}{\Delta E} \int\limits_{E_l - \frac{1}{2}\Delta E}^{E_l + \frac{1}{2}\Delta E} p(0, E) dE, \quad l = 1, \ldots, n_E. \tag{31}$$

While Eq. (30) can be solved by multiplying $\mathbf{g}$ by the computed inverse of $\mathbf{A}$, if $\mathbf{g}$ contains noisy data this may result in negative values of $p_l$ that are mathematically possible but physically unreasonable (Forney and Rothman, 2012b). Here, we find the solution that satisfies

$$\min_{\mathbf{p}} \| \mathbf{g} - \mathbf{A} \cdot \mathbf{p} \| \equiv \left[ \sum_{j=1}^{n_t} \left( g_j - \sum_{l=1}^{n_E} A_{j,l} p_j \right)^2 \right]^{\frac{1}{2}}, \tag{32}$$

subject to the constraints

$$p_l \geq 0, \quad l = 1, \ldots, n_E, \tag{33}$$

and

$$\sum_{l=1}^{n_E} p_l = 1, \quad l = 1, \ldots, n_E. \tag{34}$$

Eqs. (32)–(34) describe the model solution that minimizes the norm of the residual error (*i.e.* the root mean square error, or RMSE) while fulfilling the constraints that $\mathbf{p}$ is non-negative and sums to unity.

## 4.1 Choice of frequency factor

In order to construct the $\mathbf{A}$ matrix and solve for $\mathbf{p}$, our method requires that the Arrhenius frequency factor $\omega$ is prescribed *a priori*. There exists significant discussion in the literature on the best choice of $\omega$, as multiple values can describe laboratory results equally well but will result in drastically different predictions of OC degradation rates over geologic timescales (Braun and Burnham, 1987; Dieckmann, 2005). Furthermore, it has been argued that $\omega$ represents a variable change in entropy associated with the decay of specific organic compounds and should therefore be parameterized as a function of $E$ (the so-called "kinetic compensation effect" or KCE; Dieckmann, 2005; Lakshmanan et al., 1991; Tang et al., 2000). For example, a linear $\omega$ increase with $E$ from $\approx 10^8 \, \text{s}^{-1}$ ($E = 175 \, \text{kJ} \, \text{mol}^{-1}$) to $\approx 10^{26} \, \text{s}^{-1}$ ($E = 400 \, \text{kJ} \, \text{mol}^{-1}$) has been utilized to better predict petroleum formation rates (Dieckmann, 2005). To circumvent the issue of multiplicity, and to account for the KCE, Miura and Maki (1998) developed a method to estimate the best-fit $\omega$ for each $E$ value by comparing the shift in elution temperatures when a sample is analyzed at multiple ramp rates. However, because this approach is based on large extrapolations in $1/T$ vs. $\beta/T^2$ space, it is highly sensitive to noise in temperature and $\beta$ measurements (Burnham and Braun, 1989).





To select a best-fit $\omega$, we construct the $\mathbf{A}$ matrix for a range of $\omega$ values and determine the residual error norm between measured $G(t)/G_0$ and that predicted by the resulting $\mathbf{p}$ vector determined by Eqs. (32)–(34). We consider the KCE by calculating $\omega$ as a function of $E$ according to

$$\log_{10} \omega = (\text{KCE slope})E + (\text{KCE intercept}). \tag{35}$$

Resulting residual errors for Narayani POC using a range of KCE slopes and intercepts are shown in Fig. 5 ($\beta = 5\,°\text{C min}^{-1}$, $E$ ranging from $50\,\text{kJ mol}^{-1}$ to $350\,\text{kJ mol}^{-1}$). By setting an "acceptable" residual error norm cutoff of $\leq 10^{-4}$, it can be seen that there exist multiple KCE slope and intercept combinations that can equally fit the observed data. Additionally, we estimate the best-fit $\omega$ using a range of ramp rates ($\beta = 2\,°\text{C min}^{-1}$, $5\,°\text{C min}^{-1}$ and $10\,°\text{C min}^{-1}$) following the method of Miura and Maki (1998) (Fig. 5, white circle). While this estimate falls outside of the cutoff range, likely due to noise in temperature and

$\beta$ measurements, it results in a KCE slope near zero and suggests that $\omega$ is constant during RPO oxidation of this sample. To accurately compare RPO results between samples, we therefore select a constant $\omega$ value of $10^{10}\,\text{s}^{-1}$, well within the cutoff range, for all samples analyzed herein (Fig. 5, red star). While a different choice of $\omega$ will shift $p(0, E)$ to higher or lower absolute values of $E$, we emphasize that it will not affect the distribution of $E$ and that only relative changes in $E$ between RPO fractions should be interpreted.

For example, a shift in $\omega$ from a constant value of $10^7\,\text{s}^{-1}$ to $10^{12}\,\text{s}^{-1}$ results in an increase in the mean of the pdf of $E$, termed $\overline{E}$ and calculated as

$$\overline{E} = \sum_{l=1}^{n_E} E_l p(0, E_l)\Delta E, \tag{36}$$

from $150\,\text{kJ mol}^{-1}$ to $224\,\text{kJ mol}^{-1}$ for Narayani POC. However, the relative standard deviation of the pdf of $E$, calculated as $\sigma/\overline{E}$, where

$$\sigma^2 = \overline{E^2} - \left(\overline{E}\right)^2, \tag{37}$$

remains constant at 20%. A higher $\omega$ value therefore results in a broader $p(0, E)$ distribution that is centered at a higher mean $E$ value but has no effect on the shape of the distribution.

## 4.2   Tikhonov regularization

In principle, after choosing $\omega$ and constructing the $\mathbf{A}$ matrix, the pdf of $E$ that best describes an RPO thermogram can be

determined by solving Eqs. (32)–(34). However, the inverse solution is sensitive to noise at the level of RPO instrument precision ($\pm 5\,\text{ppm CO}_2$, $\pm 5\,°\text{C}$; Hemingway et al., 2017), and is therefore ill-posed (Hansen, 1994; Lakshmanan et al., 1991). We address this sensitivity to data uncertainty using Tikhonov regularization (Hansen, 1994; Forney and Rothman, 2012a, b).

   This approach finds an optimal solution that minimizes $p(0, E)$ complexity (as determined by the intensity of fluctuations, or "roughness") while maximizing solution accuracy. Following Forney and Rothman (2012b), we calculate roughness as the





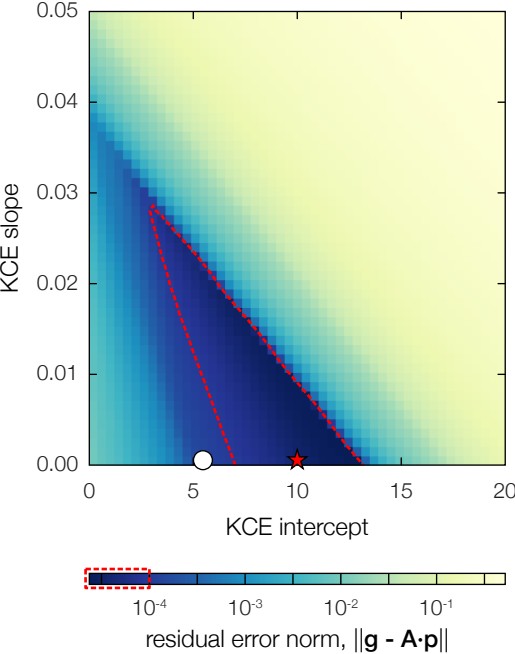

**Figure 5.** Frequency factor assessment. Model residual error norm using a range of KCE slopes and intercepts for Narayani POC ($\beta =$ $5\,°\mathrm{C\,min}^{-1}$). Each pixel represents the best-fit solution to Eqs. (32)–(34) for a given $\omega$ as determined by Eq. (35). "Acceptable" fits with residual error norm $\leq 10^{-4}$ are contained within the red dotted line. Estimated result using the method of Miura and Maki (1998) for 3 ramp rates ($\beta = 2\,°\mathrm{C\,min}^{-1}$, $5\,°\mathrm{C\,min}^{-1}$ and $10\,°\mathrm{C\,min}^{-1}$) is plotted as a white circle, while the point corresponding to $\omega = 10^{10}\,\mathrm{s}^{-1}$ (the value chosen for all samples in this study) is plotted as a red star.

first derivative of the solution vector:

$$\left\| \frac{dp(0,E)}{dE} \right\| = \left[ \sum_{l=2}^{n_E-1} \left( \frac{p_{l+1}-p_l}{\Delta E} \right)^2 \right]^{\frac{1}{2}} \equiv \| \mathbf{R} \cdot \mathbf{p} \|, \tag{38}$$

where $\mathbf{R}$ is the bi-diagonal first derivative operator with an additional first row equal to $\begin{bmatrix} 1 & 0 \end{bmatrix}$ and last row equal to $\begin{bmatrix} 0 & -1 \end{bmatrix}$ to account for $\mathbf{p}$ being equal to zero outside of the range $E_{\min} < E < E_{\max}$. The regularized inverse solution can then be 5 determined by including this roughness term when solving the constrained least squares:

$$\min_{\mathbf{p}} \| \mathbf{g} - \mathbf{A} \cdot \mathbf{p} \| + \lambda \| \mathbf{R} \cdot \mathbf{p} \|, \tag{39}$$

where $\lambda$ is a scalar that determines how much to weight the roughness $\| \mathbf{R} \cdot \mathbf{p} \|$ relative to the residual error $\| \mathbf{g} - \mathbf{A} \cdot \mathbf{p} \|$. The best choice of $\lambda$ is considered to be the value that optimizes this balance. As described in Hansen (1994), a common approach is to choose the value corresponding to the point of maximum curvature in a $\log-\log$ plot of residual error and roughness 10 while allowing $\lambda$ to range over many orders of magnitude (*i.e.* the so-called "L-curve"). From this point, increasing $\lambda$ strongly increases residual error but has little effect on solution roughness, while decreasing $\lambda$ greatly increases roughness but has little





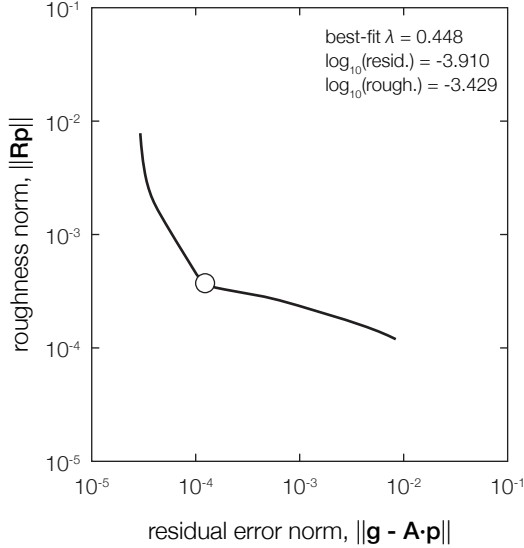

**Figure 6.** Tikhonov Regularization L-curve for Narayani POC ($\beta = 5\,°\mathrm{C\,min}^{-1}$). White circle corresponds to the point of maximum curvature (*i.e.* the best-fit $\lambda$ value).

effect on residual error. Thus, here we choose the $\lambda$ value corresponding to the corner of the L-curve for each sample, as exemplified in Fig. 6.

### 4.3  $p(0, E)$: A novel proxy for chemical bond strength

In order to interpret $p(0, E)$ as an intrinsic property of OC contained within a sample, we must show that results do not depend

5    on experimental conditions such as ramp rate $\beta$ and initial carbon mass $G_0$. To test this, we analyzed Narayani POC using a range of masses ($G_0 = 268\,\mu\mathrm{g\,C}$, $533\,\mu\mathrm{g\,C}$ and $828\,\mu\mathrm{g\,C}$) and ramp rates ($\beta = 2\,°\mathrm{C\,min}^{-1}$, $5\,°\mathrm{C\,min}^{-1}$ and $10\,°\mathrm{C\,min}^{-1}$). Fig. 7 shows that the regularized pdfs of $E$ are nearly identical across all experimental conditions. This supports the hypothesis that the $p(0, E)$ distribution is an intrinsic property of a given sample when exposed to a particular oxidation pathway. Although there exist small differences between individual analyses due to measurement uncertainty and variability in best-fit $\lambda$ values

10    (ranging from $0.044$ to $0.448$, $n = 5$), the main features of $p(0, E)$ are robust across all conditions.

We propose $p(0, E)$ as a novel proxy to describe the distribution of carbon bond strength (Braun and Burnham, 1987; Burnham and Braun, 1999; Cramer, 2004). For example, Narayani POC is known to integrate recently fixed biomass, pre-aged soils, and eroded rock-derived material (Galy et al., 2008, 2011; Galy and Eglinton, 2011; Rosenheim and Galy, 2012). Such integration should lead to large chemical diversity and a broad, complex $E$ distribution, as is observed (Fig. 7). Furthermore,

15    slow environmental turnover has been shown to diversify the distribution of chemical bonds due to a combination of microbial alterations (Schmidt et al., 2011), OC aggregation (Keil and Mayer, 2014), and stabilization by mineral surfaces (Keil and





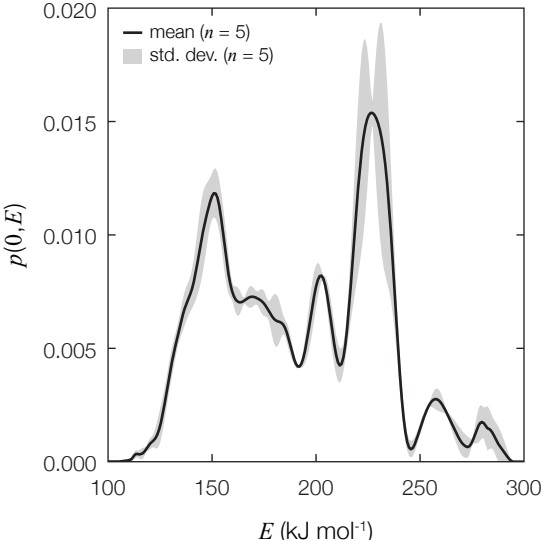

**Figure 7.** Regularized $p(0, E)$ distribution for Narayani POC. Mean (black line) and standard deviation (gray shaded region) of $p(0, E)$ analyzed using a range of $G_0$ and $\beta$ values ($n = 5$). Narrow standard deviation indicates that model results are independent of experimental conditions.

Mayer, 2014; Mikutta et al., 2006). Thus, OC reactivity within the RPO instrument and the resulting $E$ distribution likely reflects both the strength of covalent bonds between carbon atoms as well as interactions with mineral surfaces (Keil and Mayer, 2014; Mikutta et al., 2006). We therefore propose that combining $p(0, E)$ with serial oxidation isotope measurements is an ideal method to test the effects of mineral interactions and selective preservation on OC turnover time.

## 5 Relating $E$ and isotope composition

### 5.1 Determining the distribution of $E$ within each RPO fraction

To relate $p(0, E)$ distributions to RPO isotope measurements, we calculate the subset of the pdf of $E$ that is contained within each RPO fraction. Because we can predict the evolution of $p(t, E)$ at any time $t$ following Eq. (21), this can be calculated as

$$\Pi_f(E) = p(t_1, E) - p(t_2, E), \quad f = 1, \ldots, n_f, \tag{40}$$

where $n_f$ is the number of RPO fractions collected for a given sample, $\Pi_f(E)$ is the subset of $p(0, E)$ contained in RPO fraction $f$, and $t_1$ and $t_2$ are the initial and final time points, respectively, of $CO_2$ collection for RPO fraction $f$. Resulting $\Pi_f(E)$ distributions for Narayani POC are shown in Fig. 8. Finally, in order to generate $E$ vs. $\delta^{13}C$ and $E$ vs. Fm scatter plots,



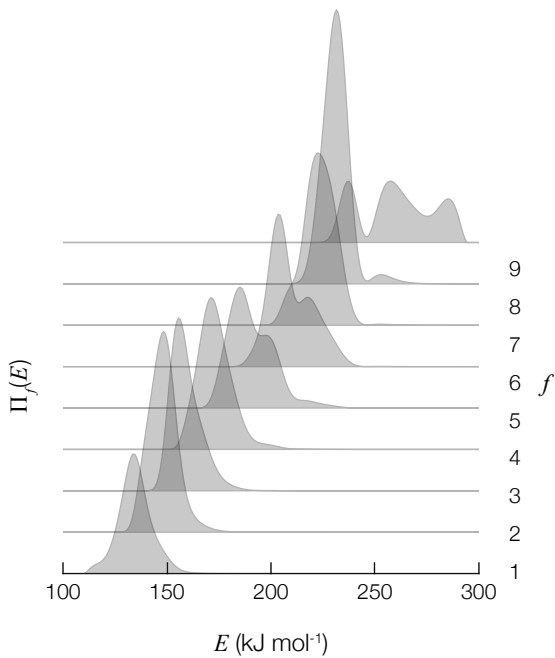

**Figure 8.** $\Pi_f(E)$ distributions for Narayani POC ($f = 1, \ldots, 9$). Each $\Pi_f(E)$ represents the range of $E$ values contained within RPO fraction $f$. The sum of all $\Pi_f(E)$ distributions shown here thus yields the $p(0, E)$ distribution shown in Fig. 7. Distributions have been staggered along the $y$ axis for visual clarity. $\Pi_f(E)$ distributions do not follow any predictable functional form and are highly overlapping due to the fact that OC associated with a given $E$ value decays over a wide time interval (Cramer, 2004).

we calculate the mean $E$ value contained in each RPO fraction as

$$\overline{E}_f = \sum_{l=1}^{n_E} E_l \Pi_f(E_l) \Delta E, \quad f = 1, \ldots, n_f \tag{41}$$

and the standard deviation of $E$ contained in each RPO fraction as $\sigma_f$, where

$$\sigma_f^2 = \overline{E^2}_f - \left(\overline{E}_f\right)^2, \quad f = 1, \ldots, n_f. \tag{42}$$

5  Resulting $\overline{E}_f$ and $\sigma_f$ values are reported in Table 2. It can be seen in Fig. 8 that $\Pi_f(E)$ distributions do not follow any particular form (*e.g.* Gaussian) and are highly overlapping, reflecting the fact that the $CO_2$ isotope composition for each RPO fraction is itself a weighted average of multiple sources.

## 5.2  Kinetic isotope fractionation

While not necessary for Fm because it is fractionation-corrected by definition (Reimer et al., 2004), it is important to correct for
10  any kinetic isotope effects occurring within the RPO instrument before interpreting $\delta^{13}C$ as a carbon source tracer (Hemingway et al., 2017). If kinetic fractionation is large, as has been observed both during thermogenic methane formation (Tang et al.,




2000; Cramer, 2004) and dissolved OC oxidation by *uv* light (Oba and Naraoka, 2008), then this effect could overprint carbon source $\delta^{13}$C signals. However, when directly measured using single-compound standards, Hemingway et al. (2017) concluded that $^{13}$C fractionation within the RPO instrument must be $\leq 2‰$. Still, we correct the measured $\delta^{13}$C values of each RPO fraction using the ratio of carbon-normalized $^{13}$C and $^{12}$C decomposition rates at each time point according to

$$^{13/12}\mathrm{r}(t) = \frac{\left(\frac{d^{13}\mathrm{G}(t)}{dt}\right)}{\left(\frac{d^{12}\mathrm{G}(t)}{dt}\right)} \left(\frac{^{12}\mathrm{G}_0}{^{13}\mathrm{G}_0}\right), \tag{43}$$

where we have added a preceding 12 or 13 superscript to specify isotope-specific variables. Following the Arrhenius equation, $^{13/12}\mathrm{r}(t)$ can be written as a function of the difference in $E$ between $^{13}$C- and $^{12}$C-containing molecules:

$$^{13-12}\Delta\mathrm{E} = {}^{13}\mathrm{E} - {}^{12}\mathrm{E}. \tag{44}$$

Although $^{13-12}\Delta\mathrm{E}$ is likely not identical for all compounds due to differences in the entropy and enthalpy of isotope substitu-
tion (Tang et al., 2000), the estimated range of values for RPO analysis is small ($0.3 \times 10^{-3}\,\mathrm{kJ\,mol^{-1}}$ to $1.8 \times 10^{-3}\,\mathrm{kJ\,mol^{-1}}$; Hemingway et al., 2017). We therefore assume a $^{13-12}\Delta\mathrm{E}$ value of $1.8 \times 10^{-3}\,\mathrm{kJ\,mol^{-1}}$ for all RPO fractions, noting that a choice of $0.3 \times 10^{-3}\,\mathrm{kJ\,mol^{-1}}$ would result in $\delta^{13}$C values that are identical to those calculated here within analytical uncertainty.

Values of $^{13/12}\mathrm{r}(t)$ can be determined using the ratio of carbon-normalized, isotope-specific decay rates by substituting
$p(0, {}^{12}\mathrm{E})$ and $p(0, {}^{13}\mathrm{E})$ for $p(0,E)$ in Eq. (19). Because carbon is present as $\approx 99\,\%$ $^{12}$C, we set $p(0, {}^{12}\mathrm{E})$ equal to $p(0,E)$ such that

$$\frac{d^{12}\mathrm{G}(t)}{dt} = \frac{d\mathrm{G}(t)}{dt}. \tag{45}$$

Corresponding $d^{13}\mathrm{G}(t)/dt$ can then be determined using

$$p(0, {}^{13}\mathrm{E}) = p(0, E + {}^{13-12}\Delta\mathrm{E}). \tag{46}$$

$^{13}$C-containing molecules decay at rates governed by a pdf of $E$ that is identical to $p(0,E)$ but has been shifted by $1.8 \times 10^{-3}$ kJ mol$^{-1}$. We then correct the measured $\delta^{13}$C values of each RPO fraction $f$ according to

$$\delta^{13}\mathrm{C}_f^{\mathrm{corrected}} =$$
$$\frac{1}{^{13/12}\mathrm{r}(t)_f^{\mathrm{av}}} \left(\delta^{13}\mathrm{C}_f + 1000\left[{}^{13/12}\mathrm{r}(t)_f^{\mathrm{av}} - 1\right]\right),$$
$$f = 1, \ldots, n_f, \tag{47}$$

where $^{13/12}\mathrm{r}(t)_f^{\mathrm{av}}$ is the average $^{13/12}\mathrm{r}(t)_f$ value over the time of collection for RPO fraction $f$. For the samples analyzed here,
$^{13/12}\mathrm{r}(t)$ is initially $\approx 0.999$, indicating slightly faster $^{12}$C decay at low temperatures, and gradually increases to $\approx 1.002$ when $G(t) \ll 0.01G_0$, as has been described previously (Cramer, 2004; Hemingway et al., 2017). Resulting kinetic fractionation corrections are near or within analytical uncertainty, with absolute $\delta^{13}$C values for all RPO fractions shifted by $<0.2‰$.





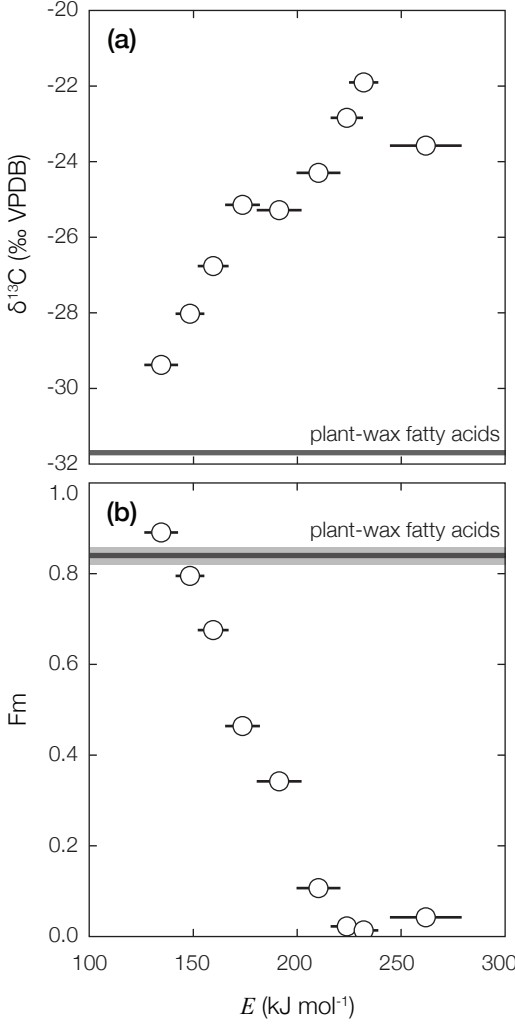

**Figure 9.** $E$ vs. isotope relationships. **(a)** $E$ vs. $\delta^{13}$C and **(b)** $E$ vs. Fm for Narayani POC. All isotope values have been corrected for blank carbon contribution following Hemingway et al. (2017), and $\delta^{13}$C values have additionally been corrected for kinetic fractionation. Gray lines and shading are the plant-wax fatty acid biomarker isotope values (mean $\pm 1$ std. dev. analytical uncertainty; Galy et al., 2011; Galy and Eglinton, 2011). Note that plant-wax fatty acids are known to contain less $^{13}$C (lower $\delta^{13}$C values) than corresponding biospheric OC. Each point is plotted at $E = \overline{E}_f$. Error bars in $E$ are equal to $\sigma_f$, while $\delta^{13}$C and Fm analytical uncertainty is always smaller than point marker and is therefore not shown.

## 5.3 Comparing $E$ to $\delta^{13}$C and Fm

Finally, we describe a framework to directly relate OC reactivity and isotope distributions by plotting $\overline{E}_f$ for each RPO fraction vs. the corresponding measured $\delta^{13}$C and Fm values (Table 2). Resulting relationships, as well as plant-wax fatty



acid isotope values (Galy et al., 2011; Galy and Eglinton, 2011), are shown in Fig. 9. Within this framework, it can be seen that Narayani POC must contain at least two end members with drastically different isotope compositions and unique yet overlapping $E$ distributions. Previous studies have shown that $\approx (20 \pm 5)\,\%$ of OC contained in this sample is derived from the erosion of carbon-rich bedrock (Galy et al., 2008; Rosenheim and Galy, 2012). Rock-derived OC is the likeliest source of

high-$E$, low-Fm material, as this end member is $^{14}$C-free by definition. Plant-wax FA $\delta^{13}$C and Fm values are similar to those for low-$E$ RPO fractions (Fig. 9), suggesting that vascular plant OC is the source of low-$E$ material. Narayani POC isotope trends are thus consistent with predominantly biospheric carbon below $\approx 150\,\mathrm{kJ\,mol}^{-1}$, a mixed region from $\approx 150\,\mathrm{kJ\,mol}^{-1}$ to $\approx 200\,\mathrm{kJ\,mol}^{-1}$, and exclusively rock-derived OC above $\approx 200\,\mathrm{kJ\,mol}^{-1}$. This result exemplifies the utility of RPO $E$ vs. isotope relationships to directly relate the distribution of OC sources, environmental turnover times, and chemical bonding

environments.

## 6   Conclusions

In this study, we present a regularized, inverse method to determine the distribution of $E$, a measure of OC reactivity, when natural organic matter is exposed to serial oxidation. We show that OC decay follows parallel, first-order kinetics. In contrast, the kinetics of carbonate oxidation cannot be constrained due to matrix effects. We propose that $p(0, E)$, the distribution of

$E$ contained within a sample, is a useful proxy to describe the range of OC bonding environments. Importantly, our method does not require *a priori* assumptions about the distributional form of $p(0, E)$. Finally, we determine the subset of $E$ contained within each RPO fraction in order to directly relate reaction energetics with the distribution of carbon isotope compositions within a complex OC mixture. We suggest that $E$ vs. isotope relationships can provide new insight into understanding the compositional controls on OC source and residence time. This manuscript is accompanied by an open-source Python package

for performing all analyses.

*Code and data availability.*  All thermogram data are available in the supplementary material. The open-source 'rampedpyrox' package is accessible using the Python Package Index (http://pypi.python.org/pypi/rampedpyrox).

*Author contributions.*  J.D.H., S.Z.R., and V.V.G. performed all laboratory measurements; J.D.H., V.V.G., and D.H.R. analyzed the data; J.D.H. and D.H.R. developed the inverse model; J.D.H. wrote the manuscript with input from all authors.

*Competing interests.*  The authors declare that they have no conflict of interest.





*Acknowledgements.* We thank the entire NOSAMS facility, especially Ann McNichol, Al Gagnon, Steven Beaupré, and Mary Lardie, for assistance with the RPO instrument. This research was supported by: the NSF Graduate Research Fellowship Program grant number 2012126152 (J.D.H.); NASA Astrobiology grant number NNA13AA90A and NSF grant number EAR-1338810 (D.H.R); and the WHOI Independent Study Award (V.V.G.).



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



**Table 1.** List of mathematical symbols used throughout this study.

| Symbol | Parameter | Units |
|---|---|---|
| $\mathbf{A}$ | Dynamic disordered kinetic design matrix | $\mathrm{kJ\,mol^{-1}}$ |
| $\alpha(t)$ | Integral of $G_0$-normalized thermogram at time $t$ | – |
| $\beta$ | Temperature ramp rate | $\mathrm{K\,s^{-1}}$ |
| $\delta^{13}\mathrm{C}_f$ | $^{13}\mathrm{C}/^{12}\mathrm{C}$ ratio of RPO fraction $f$, expressed in per mille VPDB | ‰ |
| $\Delta E$ | Activation energy step | $\mathrm{kJ\,mol^{-1}}$ |
| $\Delta t_j$ | Time step for point $j$ in $\mathbf{t}$ | s |
| $^{13-12}\Delta\mathrm{E}$ | $E$ difference between $^{13}\mathrm{C}$- and $^{12}\mathrm{C}$-containing compounds | $\mathrm{kJ\,mol^{-1}}$ |
| $E_i$ | Activation energy for component $i$ | $\mathrm{kJ\,mol^{-1}}$ |
| $E$ | Continuous form of $E_i$ | $\mathrm{kJ\,mol^{-1}}$ |
| $\mathbf{E}$ | Vector of discretized activation energy | $\mathrm{kJ\,mol^{-1}}$ |
| $\mathrm{Fm}_f$ | $^{14}\mathrm{C}/^{12}\mathrm{C}$ ratio for RPO fraction $f$, expressed as fraction modern | – |
| $G_0$ | Total initial mass of carbon | µg C |
| $g_i(0)$ | Initial mass of carbon in component $i$ | µg C |
| $g_i(t)$ | Mass of carbon in component $i$ remaining at time $t$ | µg C |
| $G(t)$ | Mass of total carbon remaining at time $t$ | µg C |
| $g(0, E)$ | Continuous form of $g_i(0)$ | µg C |
| $g(t, E)$ | Continuous form of $g_i(t)$ | µg C |
| $\mathbf{g}$ | Vector of $G(t)/G_0$ at each time point | – |
| $k_i(t)$ | First-order rate coefficient for component $i$ at time $t$ | $\mathrm{s^{-1}}$ |
| $k(t, E)$ | Continuous first-order rate coefficient for energy value $E$ at time $t$ | $\mathrm{s^{-1}}$ |
| $\kappa_i(t)$ | Discrete, time-integrated first-order decay coefficient for component $i$ at time $t$ | – |
| $\kappa(t, E)$ | Continuous, time-integrated first-order decay coefficient for energy value $E$ at time $t$ | – |
| $\lambda$ | Regularization weighting factor | – |
| $\mathrm{m}_f$ | Mass of carbon (as $CO_2$) contained in RPO fraction $f$ | µg C |
| $m(t)$ | $G_0$-normalized decay rate at time $t$ | $\mathrm{s^{-1}}$ |
| $n_E$ | Number of nodes in $\mathbf{E}$ | – |
| $n_t$ | Number of nodes in $\mathbf{t}$ | – |
| $p_i(0)$ | Fraction of $G_0$ initially in component $i$ | – |
| $p_i(t)$ | Fraction of $G_0$ remaining in component $i$ at time $t$ | – |
| $p(0, E)$ | Continuous form of $p_i(0)$ | – |
| $p(t, E)$ | Continuous form of $p_i(t)$ | – |
| $\mathbf{p}$ | Vector of $p(0, E)/\Delta E$ at each energy point | $(\mathrm{kJ\,mol^{-1}})^{-1}$ |
| $\Pi_f(E)$ | Subset of $p(0, E)$ contained in RPO fraction $f$ | – |
| $^{13/12}\mathrm{r}(t)$ | Ratio of $^{13}\mathrm{C}/^{12}\mathrm{C}$ decay at time $t$ | – |
| $R$ | Ideal gas constant | $\mathrm{kJ\,mol^{-1}\,K^{-1}}$ |
| $\mathbf{R}$ | First derivative operator matrix | – |
| $T(t)$ | Temperature at time $t$ | K |
| $\mathbf{t}$ | Vector of discretized time | s |
| $\omega$ | Arrhenius pre-exponential ("frequency") factor | $\mathrm{s^{-1}}$ |





**Table 2.** Narayani POC RPO temperature ranges, carbon masses, $\delta^{13}$C, Fm, and $E$ for each fraction, $f$. All masses and isotope values are blank corrected following Hemingway et al. (2017). See Eqs. (41)–(42) for $E$ calcuations.

| $f$ | $T$ (°C) | | $m_f$ ($\mu$g C) | | $\delta^{13}$C$_f$ (‰ VPDB)* | | Fm$_f$ | | $E$ (kJ mol$^{-1}$)** | |
|---|---|---|---|---|---|---|---|---|---|---|
| | min. | max. | mean | std. dev. | mean | std. dev. | mean | std. dev. | $\overline{E}_f$ | $\sigma_f$ |
| 1 | 150 | 310 | 68.4 | 0.7 | -29.5 | 0.2 | 0.891 | 0.004 | 134.4 | 8.1 |
| 2 | 310 | 367 | 105.6 | 1.1 | -28.1 | 0.2 | 0.795 | 0.002 | 147.9 | 7.1 |
| 3 | 367 | 412 | 82.4 | 0.8 | -26.7 | 0.2 | 0.676 | 0.003 | 159.0 | 7.5 |
| 4 | 412 | 475 | 92.6 | 0.9 | -25.1 | 0.2 | 0.464 | 0.003 | 173.1 | 8.5 |
| 5 | 475 | 545 | 85.6 | 0.9 | -25.3 | 0.2 | 0.342 | 0.003 | 190.6 | 10.9 |
| 6 | 545 | 610 | 98.4 | 1.0 | -24.3 | 0.2 | 0.107 | 0.002 | 209.7 | 10.7 |
| 7 | 610 | 661 | 101.5 | 1.0 | -22.9 | 0.2 | 0.022 | 0.002 | 223.4 | 8.0 |
| 8 | 664 | 725 | 125.6 | 1.3 | -21.8 | 0.2 | 0.014 | 0.002 | 231.5 | 7.1 |
| 9 | 725 | 997 | 86.6 | 0.9 | -23.5 | 0.2 | 0.042 | 0.002 | 260.5 | 17.7 |

*$\delta^{13}$C$_f$ is additionally corrected following Hemingway et al. (2017) to ensure that the mass-weighted mean matches the measured bulk value.

**Assuming L-curve best-fit $\lambda$ value and $\omega = 10 \times 10^{10}$ s$^{-1}$.

**Table 3.** JGOFS RPO temperature ranges, carbon masses, and $\delta^{13}$C for each fraction, $f$. All masses and isotope values are blank corrected following Hemingway et al. (2017).

| $f$ | $T$ (°C) | | $m_f$ ($\mu$g C) | | $\delta^{13}$C$_f$ (‰ VPDB)* | |
|---|---|---|---|---|---|---|
| | min. | max. | mean | std. dev. | mean | std. dev. |
| 1 | 163 | 363 | 38.5 | 0.4 | -20.1 | 0.2 |
| 2 | 363 | 435 | 45.9 | 0.5 | -10.3 | 0.2 |
| 3 | 435 | 543 | 217.6 | 2.2 | -0.4 | 0.2 |
| 4 | 543 | 597 | 154.4 | 1.5 | 0.3 | 0.2 |
| 5 | 597 | 720 | 497.7 | 5.0 | 0.9 | 0.2 |

*$\delta^{13}$C$_f$ is additionally corrected following Hemingway et al. (2017) to ensure that the mass-weighted mean matches the measured bulk value.