# Peer review of "Technical note: An inverse method to relate organic carbon reactivity to isotope composition from serial oxidation"

_Biogeosciences, 2017_

## Referee Comment (RC1) · D. Burdige (Referee) · 1 Sep 2017

This manuscript describes a mathematical technique for the analysis of serial oxidation results (i.e., thermograms), which effectively allows one to convert the temperature of oxidation on the x-axis of a thermogram to an activation energy (also on the x-axis) for the oxidation of organic matter occurring at that temperature. When combined with isotopic measurements of the $CO_2$ being produced by the oxidation, the authors suggest that this can be used to infer information about the reactivity of organic matter in sediments.

I think that the results presented here for the one sample that was analyzed are in-

teresting and provide intriguing results. Some of the proposed uses of this approach (e.g., p. 17 line 4 p. 21 line 8) may indeed turn out to be correct, but these conclusions may also be a bit premature based on the information presented here.

Related to this I am concerned about the "validation" of the approach based on the analysis of a single suspended sediment sample, in part because little is presented in this analysis to independently verify the results. It seems to me that one way to verify this approach would involve taking well-defined organic compounds whose activation energy for oxidation is either known or can be estimated, subjecting them to ramped pyrolysis/oxidation and seeing if the activation energies that the analysis of these thermograms gives back agrees with these values. To be honest, I'm not sure how well-known or easy it is to obtain the activation energy for the oxidation of single organic compounds. However, a quick Google search of "activation energy oxidation of organic compounds" yielded what looked like a number of promising hits.

At the same time, I think there is actually some data in the literature that could be used in some simple, albeit qualitative, verification of the results discussed in section 5.3. For example, Westrich and Berner (1988) suggest, at least in the coastal sediments they studied, that organic matter which is less susceptible to decomposition may have a higher activation energy for decomposition (as one might infer from the results here in section 5.3). Similar observation are also presented in Middelburg et al. (1996). Although I'm not familiar with the papers cited on p. 12 lines 17-18, I also wonder whether information in these papers might be useful here as well.

In general, the presentation of the method is rather dense in places and there are several places where I found things confusing and/or where more information about the mathematical derivation is needed. Note that (x,y) refers to page x, line y.

1. Starting near the bottom of p. 6 (line 23) "thermograms" and "mass-normalized decay rates" seemed to be used somewhat interchangeably (also see the captions for Fig. 2-4). However, there was not a clear explanation (at least that I could find) about

why this is the case. This may need to be clarified. Addressing this question should also help explain why the y-axes in Figs. 2, 3 and 4(b) all have different units.

2. (9, 20) - It was not clear which model is being referred to here by "our" model.

3. (10,19) – What are "short-range-order" minerals?

4. (10, 22) – Is there are reason why here and in Fig. 4 the x-axis has changed from temperature to time?

5. (13, 6) – Should it say "Eq. (30) can be solved for p by multiplying . . ."?

6. (13, 8) - How exactly do you find the solution (i.e., the p vector) that satisfies Eq. (32)?

7. (14, 7-9) – How exactly does the method of Miura and Maki (1998) differ from that used here?

8. (14, 21-22) – If a higher $\omega$ value results in a broader p(0,E) how can it also have "no effect on the shape of the distribution"? What am I missing?

9. (15, 3) – What are the dimensions of R, and do the bold 0's in the description of the first and last rows of this matrix mean that all of the other values in the row are 0? If this is so, other may also not be familiar with this notation, and I think this could be made a little clearer.

10. (15, 5) – What is meant here by "solving the constrained least squares"? I kept thinking this was similar in someway to how Eq. (32) was used to solve for the p vector, but Eq. (39) just didn't make sense to me in that way. Again, am I missing something here?

11. It's also not clear to me how Fig. 6 was generated, and the interpretation of this figure starting on line 7, p. 15.

12. (16,15) – What exactly is meant by "diversify the distribution of chemical bonds"?

13. The general shape of the thermogram in Fig. 2 looks awfully similar to the p(0,E) distribution in Fig. 7. Does that means that activation energy scales linearly (more or less) with temperature of pyrolysis? In general that makes intuitive sense, and this is also discussed here briefly on p. 9, line 15. It will be interesting to see if this is a general trend observed across a broader range of samples.

References cited

Middelburg, J. J., G. Klaver, J. Nieuwenhuize, A. Wilemake, W. de Hass, T. Vlug, and J. F. W. A. van der Nat. 1996. Organic matter mineralization in intertidal sediments along an estuarine gradient. Mar. Ecol. Prog. Ser. 132, 157-168.

Westrich, J. T., and R. A. Berner. 1988. The effect of temperature on rates of sulfate reduction in marine sediments. Geomicrobiol. J. 6, 99-117.

---

## Referee Comment (RC2) · B. Boudreau (Referee) · 3 Sep 2017

Hemingway et al. have offered an inverse model to extract the reactivity of organic matter (OM) and relate it to the isotopic composition via data obtained from a record of thermal decomposition of that organic matter. This is a technique that has been used with respect to petroleum formation and there exists published literature for that application. The present paper hopes to extend the method to the degradation of OM sampled in low-temperature aquatic systems.

The mathematics of the model appear to be solid (better than my own efforts in this area), including the use of a Lagrange multiplier to add constraints to the model solution. I do not think that the paper can be faulted on this account; nevertheless, I have two strong reservations.

Firstly, microbial enzymatic degradation of OM is not the same process as thermal decomposition. Microbes use enzymes to breakdown OM in order to increase the rate of this reaction. According to a very broad interpretation of transition-state theory, that result is obtained by altering the decay (reaction) mechanism so as to lower the activation energy of the reaction. The authors' thermal method is also based on activation energy, but on the activation energy for a thermal decomposition reaction. Thus the microbial and the thermal activation energies are not guaranteed to be the same or even comparable. Assigning the thermally derived results to the microbial situation is not, at this time, experimentally justified.

Secondly, only two samples were tested with the method, and one, OM from a marine sediment, failed the test of the model assumptions. I am unaware of any other papers that have applied this technique to aquatic "low-temperature" sediments. That represents meager testing of the applicability of the model. The low-temperature geochemical community does not at this stage know if the method is useful, and the theory has significantly overstepped the acceptance of the methodology itself.

---

## Author Comment (AC1) · 21 Sep 2017

Prof. Burdige,

We thank you for your helpful concerns and feedbacks on our manuscript. Below you will find our response to your comments, along with some corresponding changes that we will make to the text once the discussion period ends. We invite further dialogue if anything is unclear or if you believe more explanation is required.

**This manuscript describes a mathematical technique for the analysis of serial oxidation results (i.e., thermograms), which effectively allows one to convert the**

[Figure]

**temperature of oxidation on the x-axis of a thermogram to an activation energy (also on the x-axis) for the oxidation of organic matter occurring at that temperature. When combined with isotopic measurements of the $CO_2$ being produced by the oxidation, the authors suggest that this can be used to infer information about the reactivity of organic matter in sediments.**

RE: We would like to clarify an important point (also see our response to Reviewer 2): we do not suggest that *thermal* reactivity measured here is equivalent to, or even necessarily scales with, *microbial* reactivity within sediments. Rather, we present thermal $E$ distributions as a proxy for the distribution of chemical bonding environments. Then, by comparing $E$ distributions and corresponding isotope compositions between multiple environmental samples, we propose that this method is able to probe how, if at all, chemical composition controls OC turnover time. For example, it is entirely possible that OC described by high thermal $E$ (likely condensed, aromatic material) is consumed rapidly in certain environmental settings. Using our method, this would result in a high Fm value for this material (if it is constantly replaced by "new" material with the same chemical structure that is enriched in $^{14}C$) and/or a rapid drop in the fractional contribution of this material in older samples (if no replacement occurs). We *do not* suggest that OC with high thermal $E$ will inherently be described by a slow turnover time in the environment (and thus a low Fm value).

To emphasize this point, we add the following paragraphs starting on P2, L33:

"We note that the modeling approach developed here is broadly applicable to any serial oxidation technique, although the resulting $E$ distributions will differ depending on oxidation pathway. For example, aromatic compounds such as lignin have been shown to be highly photoreactive (Spencer et al., 2009) despite their relatively high thermal recalcitrance (Williams et al., 2014) and will likely be described by lower $E$ values when oxidized with *uv* light relative to PRO analysis. Here, we choose RPO because analysis is rapid ($\approx 3$ hours per sample), requires little material ($\approx 150$ to $250$ $\mu$g C), contains minimal preparation steps, and results in small kinetic isotope fractionation

(Hemingway et al., 2017).

We therefore treat $E$ as a proxy for OC chemical structure and emphasize that thermal reactivity is not equivalent to microbial reactivity in the environment (Leifeld and von Lützow, 2014). Rather, by comparing $E$ profiles and corresponding isotope compositions across environmental samples or experimental conditions (e.g. before and after microbial degradation), our method provides a framework to probe how, if at all, OC source and turnover time (as measured by $\delta^{13}$C and Fm) is related to its chemical composition (as predicted by thermal $E$ distributions)."

**I think that the results presented here for the one sample that was analyzed are interesting and provide intriguing results. Some of the proposed uses of this approach (e.g., p. 17 line 4 p. 21 line 8) may indeed turn out to be correct, but these conclusions may also be a bit premature based on the information presented here.**

RE: Thank you for the positive review of our approach. With regards to the two specific proposed uses, we will change the language in the modified draft to read:

P17 L3-4: "We therefore propose combining $p(0, E)$ with serial oxidation isotope measurements to test the effects of..." By removing "is an ideal method to," this should remove any perceived speculation.

P21 L8: "This result provides initial evidence for the utility of RPO $E$ vs. isotope relationships..." Again, this rephrasing should remove any speculation, and more explicitly acknowledges that the results in the manuscript are indeed initial.

**Related to this I am concerned about the "validation" of the approach based on the analysis of a single suspended sediment sample, in part because little is presented in this analysis to independently verify the results.**

RE: We agree that analyzing a single "test" sample is not entirely satisfactory. However, we believe that inclusion of additional samples is beyond the scope of this technical manuscript. As presented, central focus of this manuscript is the mathematical derivation of the inverse distributed activation energy model, not the interpretation of any given sample within the global carbon cycle. To include additional samples, and to properly discuss and interpret their results within a geochemical context, would lengthen the manuscript considerably and, in our opinion, would detract from this central focus. We therefore leave the analysis of large sample sets, and the corresponding geochemical interpretation, to a companion manuscript that is currently in preparation.

**It seems to me that one way to verify this approach would involve taking well-defined organic compounds whose activation energy for oxidation is either known or can be estimated, subjecting them to ramped pyrolysis/oxidation and seeing if the activation energies that the analysis of these thermograms gives back agrees with these values. To be honest, I'm not sure how well-known or easy it is to obtain the activation energy for the oxidation of single organic compounds. However, a quick Google search of "activation energy oxidation of organic compounds" yielded what looked like a number of promising hits.**

RE: We agree that, in theory, analyzing single organic compounds and verifying the resulting $E$ values would be an ideal check of our model. In fact, we have analyzed several individual compounds (cellulose, $n$-C$_{30}$ alkane, calcite). However, this is significantly more challenging in practice. The challenge arises from the fact that each carbon atom within any compound experiences a unique bonding environment and will be described by a unique activation energy when exposed to thermal analysis. Any single compound therefore does not result in a thermogram with a single peak (and thus a single $E$ value), but rather a complex distribution (see, for example, results from pure cellulose in Williams et al., 2014).

Additionally, any calculated / estimated $E$ value will be specific to one oxidation pathway. Literature values for decay by, for example, *uv* light will therefore not be applicable to the thermal decay presented here. While there exist literature thermal $E$ values for specific compounds, all of the studies that we have encountered use similar methods

to those presented here (i.e. they are experimental rather than theoretical values) and results are highly variable (e.g. cellulose ranging from $150$ to $250$ kJ mol$^{-1}$; see references compiled in Williams et al., 2014).

**At the same time, I think there is actually some data in the literature that could be used in some simple, albeit qualitative, verification of the results discussed in section 5.3. For example, Westrich and Berner (1988) suggest, at least in the coastal sediments they studied, that organic matter which is less susceptible to decomposition may have a higher activation energy for decomposition (as one might infer from the results here in section 5.3). Similar observations are also presented in Middelburg et al. (1996). Although I'm not familiar with the papers cited on p. 12 lines 17-18, I also wonder whether information in these papers might be useful here as well.**

RE: For the reasons outlined above, our focus is to derive a method to compare the distribution of $E$ between samples rather than to interpret the absolute values of $E$. Again, we emphasize that thermal $E$ distributions are not necessarily predictive of microbial $E$, but rather serve as a proxy for the differences in OC chemical structure between samples. Because of this point, we believe that any discussion including comparisons to OC decay in environmental samples would be misleading within the present study.

That said, the literature mentioned by the reviewer does pose an intriguing question: how is thermal $E$ related to microbial decay? Based on the results of Westrich and Berner (1988), Middelburg et al. (1996), and this study (albeit with a single sample), one might infer that they are indeed correlated. However, any such relationships are certainly speculative at this time and are outside the scope of our present manuscript.

**In general, the presentation of the method is rather dense in places and there are several places where I found things confusing and/or where more information about the mathematical derivation is needed. Note that (x,y) refers to page x,**

**line y.**

**1. Starting near the bottom of p. 6 (line 23) "thermograms" and "mass-normalized decay rates" seemed to be used somewhat interchangeably (also see the captions for Fig. 2-4). However, there was not a clear explanation (at least that I could find) about why this is the case. This may need to be clarified. Addressing this question should also help explain why the y-axes in Figs. 2, 3 and 4(b) all have different units.**

RE: We apologize for the confusion – "thermograms" refers to the measured data (i.e. with units of ppm $CO_2$), while "mass-normalized decay rates" refers to thermograms that have been normalized by the initial amount of OC loaded into the system, $G_0$ (i.e. they integrate to unity). For Fig. 3, thermograms have additionally been normalized by the ramp rate, $\beta$, in order to properly compare between different ramp rates.

To avoid confusion, we will make the following changes in the updated manuscript:

P5 L 6: We will add the following sentences: "At each time point, the measured thermogram (in units of ppm $CO_2$) can be converted to an instantaneous OC decay rate (in units of $\mu gC\ s^{-1}$) using the measured gas flow rate and the ideal gas constant. 'Thermogram' and 'decay rate' are therefore used interchangeably throughout this manuscript."

Fig. 2 caption: "Mass-normalized thermograms (gray shaded region, unitless)" will be changed to "Measured thermograms (gray shaded region, ppm $CO_2$ axis not shown)"

Fig. 3: We will change the $y$ axis label to: "$G_0$- and $\beta$-normalized decay rate $\times 10^3$ $(^\circ C^{-1})$" for clarity.

Fig. 4 (Also see response to reviewer's point 4 below): We will change the y axis labels in panels (b) and (d) to: "$G_0$-normalized decay rate, $m(t) \times 10^4\ (s^{-1})$" for clarity.

**2. (9, 20) - It was not clear which model is being referred to here by "our" model.**

RE: We will replace "our model" with "the distributed activation energy model."

**3. (10,19) – What are "short-range-order" minerals?**

RE: "Short-range-order" is a term frequently used in the soil sciences community to refer to the crystalline state of specific minerals (e.g. allophane and ferrihydrite). To avoid confusion, we will change "short-range-order minerals" to "clay minerals" in the updated manuscript.

**4. (10, 22) – Is there are reason why here and in Fig. 4 the x-axis has changed from temperature to time?**

RE: Yes, the $x$ axis has changed in Fig. 4 from temperature to time because the test for first-order kinetics requires the time derivative of the amount of OC remaining, i.e. $dG(t)/dt$. Fig. 4 panels (a) and (c) are then the results of the test for first-order [Eq. (25)], shown in graphical form. While Fig. 4 panels (b) and (d) could be plotted with temperature on the $x$ axis, this would make their connection to panels (a) and (c), as well as to Eq. (25), less clear. In contrast, Fig. 2 is showing the "raw" measured data while Fig. 3 is shown specifically to illustrate the relationship between elution temperature and ramp rate as discussed on P6 L23-26. Both of these require temperature as the $x$ axis.

However, on P10 L22-24, we realize that the use of time rather than temperature could be confusing and is technically not accurate when referencing Fig. 2b. We therefore will make the following changes for clarity in the revised manuscript:

P10 L22 and L24: We will change "$t \approx 4500$ s" to "$T \approx 500°$C (corresponding to $t \approx 4500$ s)" in both instances.

Fig. 4: We will change the $y$ axis labels in panels (b) and (d) to: "$G_0$-normalized decay rate, $m(t) \times 10^4$ (s$^{-1}$)" to more explicitly connect these panels to panels (a) and (c), as well as Eq. (25).

Fig. 4 caption: We will also add the following sentence at the end of the caption to

make the connection clearer: "For each time point in panel (a), the regression slope is equivalent to $m(t)$ for that time point as shown in panel (b)."

**5. (13, 6) – Should it say "Eq. (30) can be solved for p by multiplying . . ."?**

RE: Yes, specifying that we are solving for $\mathbf{p}$ would clarify this statement. We will change this in the updated manuscript.

**6. (13, 8) - How exactly do you find the solution (i.e., the p vector) that satisfies Eq. (32)?**

RE: We will add ". . .using the non-negative least squares algorithm of Lawson and Hanson (1995) as implemented by the SciPy package for Python" after Eq. (34) to clarify how this is done.

**7. (14, 7-9) – How exactly does the method of Miura and Maki (1998) differ from that used here?**

RE: As mentioned briefly on P13 L24-26, the method of Miura and Maki (1998) involves analyzing a given sample at multiple (at least 3) ramp rates and generating a plot of $\beta/T^2$ vs. $1/T$ for each value of $\alpha$, the fraction of initial OC that has been oxidized. Because any given $\alpha$ value will occur at a slightly different $T$ for each ramp rate (e.g. Fig. 3), this will result in a straight line in $\beta/T^2$ vs. $1/T$ space for each $\alpha$ value. The slope and $y$ intercept of this line can then be used to calculate $E$ and $\omega$ values that correspond to that particular $\alpha$ value. To estimate the KCE slope and intercept, one simply generates a plot of $\beta/T^2$ vs. $1/T$ for multiple $\alpha$ values (i.e. 5 % of initial OC oxidized, 10 %, 15 %, etc.) and plots the resulting $E$ and $\omega$ estimates. However, as we mention in the text (P13 L26-27), this method requires large extrapolations and is thus subject to large uncertainty.

In contrast, our method to generate Fig. 5 requires choosing a range of $\omega$ values *a priori*, solving Eq. (32) – (34) for each value, and calculating the residual norm between the measured and predicted thermograms. These methods are quite different – ours

is a "brute force" method that does not require one to analyze a sample at multiple ramp rates. Here we simply use the method of Miura and Maki (1998) to independently justify our choice of a constant $\omega$ value (i.e. KCE slope = 0).

Because we only invoke the Miura and Maki (1998) method as an independent justification, and because they clearly outline their method within their original manuscript, we believe that further description is not necessary here. In our opinion, explaining their method in detail would only add unnecessary equations and could cause confusion.

**8. (14, 21-22) – If a higher $\omega$ value results in a broader p(0,E) how can it also have "no effect on the shape of the distribution"? What am I missing?**

RE: Perhaps this wording is confusing. What we mean is that $\omega$ is simply a scaling factor and changing its value will have no effect on the *relative* shape of the distribution, although the reviewer is certainly correct in that broadening the distribution does affect its overall shape.

As an arbitrary example, assume a sample is described by two peaks, one centered at $E = 150$ kJ/mol containing $75$ % of total OC and a second centered at $E = 200$ kJ/mol containing $25$ % of total OC. Increasing $\omega$ will increase $E$ for both peaks and will increase the width between them accordingly, but will not change the fact that there are 2 peaks and will not affect the relative peak sizes (i.e. the $75$ % and $25$ % of total OC).

We will change this line to: "...no effect on the relative shape of the distribution" in order to clarify this point.

**9. (15, 3) – What are the dimensions of R, and do the bold 0's in the description of the first and last rows of this matrix mean that all of the other values in the row are 0? If this is so, other may also not be familiar with this notation, and I think this could be made a little clearer.**

RE: $\mathbf{R}$ has dimensions $[n_E \times n_E]$ and, yes, $[\mathbf{1}0]$ refers to a row of $[1\ 0\ 0\ 0\ 0\ ...]$ and $[0\mathbf{1}]$ refers to $[...\ 0\ 0\ 0\ 1]$ as the reviewer assumes.

To avoid confusion, we will update P15 L13-14 in the revised manuscript to read:

"...where $\mathbf{R}$ is the bi-diagonal first derivative operator matrix with dimensions $[n_E \times n_E]$. To account for $\mathbf{p}$ being equal to zero outside the range $E_{min} < E < E_{max}$, we set the first and last rows of $\mathbf{R}$ to be equal to $[1\,0]$ and $[0\,1]$, respectively, where $\mathbf{0}$ refers to a zero vector of length $n_E - 1$."

**10. (15, 5) – What is meant here by "solving the constrained least squares"? I kept thinking this was similar in someway to how Eq. (32) was used to solve for the p vector, but Eq. (39) just didn't make sense to me in that way. Again, am I missing something here?**

RE: Here, "solving the constrained least squares" refers to the constraints that each value in $\mathbf{p}$ is non-negative and that $\mathbf{p}$ sums to unity [i.e. Eqs. (33) - (34)]. The reviewer is correct in thinking that Eq. (39) is analogous to Eq. (32), with the only difference being that Eq. (39) now contains a roughness term.

In an attempt to clarify this point, we will change P15 L5-7 to read:

"Similar to Eq. (32), the regularized inverse problem can then be solved for $\mathbf{p}$ by including this roughness term in the constrained least squares. That is, we solve

[Eq. (39) goes here]

for $\mathbf{p}$ subject to the constraints presented in Eqs. (33) - (34), where $\lambda$ is a scalar that determines how much to weight the roughness $||\mathbf{Rp}||$ relative to the residual error $||\mathbf{g} - \mathbf{Ap}||$."

**11. It's also not clear to me how Fig. 6 was generated, and the interpretation of this figure starting on line 7, p. 15.**

RE: Fig. 6 was generated by solving Eq. (39) for $\mathbf{p}$ using a range of possible $\lambda$ values (in this case, ranging from $\lambda = 0.001$ to $\lambda = 100$). Each $\lambda$ value will result in a unique solution for $\mathbf{p}$ that is described by a particular roughness norm and residual error norm.

[Figure]

The black line in Fig. 6 is simply the line passing through each of these solutions. As described in Hansen (1994), solutions to the bottom left of this line are outside of the possible domain (i.e. it is impossible to have a $\mathbf{p}$ vector that is both smoother and fits the data with a lower residual error) while solutions above the line represent a poor fit of the data.

At the heart of this regularization technique is determining which $\lambda$ value is deemed "best." Tikhonov regularization states that the $\lambda$ value that "best fits the data but not the noise" is the one corresponding to the point of maximum curvature in a plot of roughness vs. residual error norm (i.e. the white circle in Fig. 6). We refer the reader to Hansen (1994) and Forney and Rothman (2012b) for a detailed description and background of this technique.

To alleviate any confusion, and to refer the reader to the proper references, we will add detail to the caption of Fig. 6 in the revised manuscript to read:

"Figure 6. Tikhonov regularization L-curve for Narayani POC ($\beta = 5°$C min$^{-1}$). The black line represents the range of roughness and residual error norms that are the result of solving Eq. (39) for $\mathbf{p}$ using multiple $\lambda$ values ranging from $0.001$ to $100$. The white circle corresponds to the point of maximum curvature along this line, and is thus deemed the 'best fit' value [see Hansen (1994), Forney and Rothman (2012b) for further details on generating the L-curve and the theory behind Tikhonov regularization]."

**12. (16,15) – What exactly is meant by "diversify the distribution of chemical bonds"?**

RE: This is meant to convey the phenomenon of increasing chemical complexity of OC with increasing turnover time, as shown in the references cited in these lines. That is, interactions with particles, production of new compounds by heterotrophs, partial oxidation by *uv* light, etc. should lead to a more complex OC mixture with time than was initially present.

However, we now realize that "diversify the distribution" is probably not the best wording. We will change this to "...has been shown to enhance the diversity of chemical bonds..." in the updated manuscript.

**13. The general shape of the thermogram in Fig. 2 looks awfully similar to the $p(0,E)$ distribution in Fig. 7. Does that means that activation energy scales linearly (more or less) with temperature of pyrolysis? In general that makes intuitive sense, and this is also discussed here briefly on p. 9, line 15. It will be interesting to see if this is a general trend observed across a broader range of samples.**

RE: Yes, the general similarity is striking and does make intuitive sense. The main difference is that the thermogram shape is "smoother," while the $p(0,E)$ distribution contains more features and "sharper" peaks. This can be explained by the fact that material at a single $E$ value (for example, a delta function) will take some amount of time to fully decay and will thus decay over a wide temperature window when analyzed in the RPO instrument. (If interested, this thought experiment is shown quite nicely in Cramer, 2004). We therefore always expect that $p(0,E)$ to contain more features and sharper peaks than the corresponding thermograms, and this is indeed the case for all samples that we have analyzed thus far.

To enforce this idea, we will add the following sentences beginning on P16 L10:

"Note that the $p(0,E)$ distribution broadly resembles the initial thermogram shape (Fig. 2a and Fig. 7), albeit with more defined features and a higher roughness. This is a result of the fact that OC associated with each $E$ value will decay over a range of temperatures in the RPO instrument, thus resulting in a 'smoothed' thermogram (Cramer, 2004)."

**References cited**

**Middelburg, J. J., G. Klaver, J. Nieuwenhuize, A. Wilemake, W. de Hass, T. Vlug,**

and J. F. W. A. van der Nat. 1996. Organic matter mineralization in intertidal sediments along an estuarine gradient. Mar. Ecol. Prog. Ser. 132, 157-168.

Westrich, J. T., and R. A. Berner. 1988. The effect of temperature on rates of sulfate reduction in marine sediments. Geomicrobiol. J. 6, 99-117.

---

## Author Comment (AC2) · 21 Sep 2017

Prof. Boudreau,

We thank you for your helpful concerns and feedback on our manuscript. Below you will find our response to your two reservations, along with some corresponding changes that we will make to the text once the discussion period ends. Because your two reservations were somewhat broad and open-ended, we have attempted to address them and explain our position and reasoning as best as possible. However, we invite further dialogue if anything is unclear or if you believe more explanation is required. We thank you again for your time and feedback.

**Hemingway et al. have offered an inverse model to extract the reactivity of organic matter (OM) and relate it to the isotopic composition via data obtained from a record of thermal decomposition of that organic matter. This is a technique that has been used with respect to petroleum formation and there exists published literature for that application. The present paper hopes to extend the method to the degradation of OM sampled in low-temperature aquatic systems.**

**The mathematics of the model appear to be solid (better than my own efforts in this area), including the use of a Lagrange multiplier to add constraints to the model solution. I do not think that the paper can be faulted on this account; nevertheless, I have two strong reservations.**

**Firstly, microbial enzymatic degradation of OM is not the same process as thermal decomposition. Microbes use enzymes to breakdown OM in order to increase the rate of this reaction. According to a very broad interpretation of transition-state theory, that result is obtained by altering the decay (reaction) mechanism so as to lower the activation energy of the reaction. The authors' thermal method is also based on activation energy, but on the activation energy for a thermal decomposition reaction. Thus, the microbial and the thermal activation energies are not guaranteed to be the same or even comparable. Assigning the thermally derived results to the microbial situation is not, at this time, experimentally justified.**

RE: We thank the reviewer for allowing us to clarify this issue. While other studies have begun to compare OM thermal $E$ values to those for microbial decay, both using laboratory incubations (e.g. Leifeld and von Lützow, 2014) as well as long-term bare fallow soil experiments (e.g. Barré et al., 2016), this has never been our intention in the present study. At no point do we imply that $E$ distributions determined using the thermal analysis described here are identical, or even comparable, to those that would be obtained by serial oxidation by microbial respiration (e.g. Mahmoudi et al., 2017). In fact, to extend the reviewer's point, we expect that every oxidation pathway (microbial

respiration, thermal analysis, *uv* light, chemical hydrolysis, etc.) will involve a unique transition state intermediate and will therefore likely result in a different $E$ distribution. For example, lignin is highly thermally recalcitrant (Williams et al., 2014) yet degrades rapidly under *uv* light (Spencer et al., 2009).

Rather, here we present thermally derived $E$ as a proxy for the range of the strength of chemical bonds experienced by carbon atoms within a sample. We emphasize that this is simply a method to separate a complex OM mixture along a particular lability "axis" (i.e. thermal lability) and measure the isotope composition at multiple points along that "axis." Thermally derived $E$, along with the corresponding isotope distributions, can then be directly compared across a sample set in order to infer differences in the molecular and isotope compositions between samples. For example, the observation that our test sample approaches an Fm value of 0 at $E > 200$ kJ mol$^{-1}$ implies that this material is derived from OM-rich bedrock (see P21, L3-5), but says nothing about microbial recalcitrance (in fact, it is possible that rock-derived OM is highly bioavailable; see Petsch et al., 2001).

To emphasize this point, we will add and/or modify the following lines within the text:

1. (P1, L2-13). Throughout the abstract, we will add "thermal" before each use of the word "reactivity" in order to clarify that $E$ values calculated here apply only to thermal analysis.

2.(P2, L33). We will add the following paragraphs:

"We note that the modeling approach developed here is broadly applicable to any serial oxidation technique, although the resulting $E$ distributions will differ depending on oxidation pathway. For example, aromatic compounds such as lignin have been shown to be highly photoreactive (Spencer et al., 2009) despite their relatively high thermal recalcitrance (Williams et al., 2014) and will likely be described by lower $E$ values when oxidized with *uv* light relative to PRO analysis. Here, we choose RPO because analysis is rapid ($\approx 3$ hours per sample), requires little material ($\approx 150$ to $250\mu$g C),

contains minimal preparation steps, and results in small kinetic isotope fractionation (Hemingway et al., 2017).

We therefore treat $E$ as a proxy for OC chemical structure and emphasize that thermal reactivity is not equivalent to microbial reactivity in the environment (Leifeld and von Lützow, 2014). Rather, by comparing $E$ profiles and corresponding isotope compositions across environmental samples or experimental conditions (e.g. before and after microbial degradation), our method provides a framework to probe how, if at all, OC source and turnover time (as measured by $\delta^{13}$C and Fm) is related to its chemical composition (as predicted by thermal $E$ distributions)."

**Secondly, only two samples were tested with the method, and one, OM from a marine sediment, failed the test of the model assumptions. I am unaware of any other papers that have applied this technique to aquatic "low-temperature" sediments. That represents meager testing of the applicability of the model. The low-temperature geochemical community does not at this stage know if the method is useful, and the theory has significantly overstepped the acceptance of the methodology itself.**

RE: Again, we thank the reviewer for raising this concern. However, we disagree with the reviewer's interpretation that the marine sediment OM sample "failed the test of the model assumptions." Rather, it is the analysis of OM *combined with* inorganic carbon (IC) that failed the model assumptions. This distinction is critical. The results from this sample (e.g. Fig. 4d) emphasize the need to decarbonate sediment samples prior to RPO analysis. Because decarbonation likely alters OM composition to some degree, there exists longstanding discussion on this topic in the organic geochemistry literature at large and the RPO literature specifically (Plante et al., 2013). By including this sample within the present study, we make a kinetic argument in favor of decarbonation – that is, we show that IC decay is mass-dependent in the presence of OM and therefore does not follow first-order kinetics. This result does not imply that OM decay from this sample fails the model assumptions.

To the reviewer's larger point, we agree that additional samples would aid in solidifying the utility of the method in describing "low-temperature" OM. However, we believe that this is beyond the scope of this technical manuscript. As presented, central focus of this manuscript is the mathematical derivation of the inverse distributed activation energy model. To include additional samples, and to properly discuss and interpret their results within a geochemical context, would lengthen the manuscript considerably and, in our opinion, would detract from this central focus.

We note that this exact concern is the focus of a companion manuscript that is currently in preparation. In that publication, we subject dozens of samples to the model treatment presented here and interpret the environmental factors controlling differences in $E$ distributions. Combining these two manuscripts would result in the mathematical treatment presented here being relegated to a supplemental discussion (as was the case for the original submission of our companion manuscript, to which those reviewers and editor suggested we separate the mathematical treatment). We believe that the mathematics and theory contain adequate nuance and require sufficient discussion to warrant this technical manuscript, rather than being relegated to a supplemental discussion. Thus, we believe that inclusion of multiple samples (and the corresponding geochemical discussion) is beyond the scope of this manuscript.

In order to emphasize that the results presented in this manuscript are preliminary (i.e. based on a single sample) and form a theoretical basis for future study, we will add and/or modify the following text:

P16, L11: "While further study is required to assess the general applicability of this technique, we propose $p(0, E)$ as a novel proxy to describe the distribution of carbon bond strength."

P17 L3-4: "We therefore propose combining $p(0, E)$ with serial oxidation isotope measurements to test the effects of..." By removing "is an ideal method to," this should remove any perceived speculation.

P21 L8: "This result provides initial evidence for the utility of RPO $E$ vs. isotope relationships. . ." Again, this rephrasing should remove any speculation, and more explicitly acknowledges that the results in the manuscript are indeed initial.

P21 L19: "We suggest that $E$ vs. isotope relationships can provide new insight into understanding the compositional controls on OC source and residence time, although we note that further study is required in order to test the general applicability of this result."

---

## Referee Comment (RC3) · B. Boudreau (Referee) · 22 Sep 2017

I thank the authors for their consideration of my comments and their willingness to address them in their paper.

---

## Referee Comment (RC4) · D. Burdige (Referee) · 22 Sep 2017

I think the changes made by the authors clarify the points I raised in my review and will make the manuscript more accessible to its reader. I appreciate the time the authors spent carefully addressing my comments.

---

## Author Comment (AC3) · 5 Oct 2017

We thank both reviewers for their time and insight – their comments will certainly make this manuscript more clear and accessible to the reader.